# 131 genetic loci highlight immunological pathways and tissues in nasal polyposis and asthma

Elmo C. Saarentaus [1,2,3] ✉, Kasper Fischer-Rasmussen [4], Eeva Sliz [5,6,7], Argyro Bizaki-Vallaskangas [8,9], FinnGen*, Tarja Laitinen[2], Sanna Toppila-Salmi[10,11,12], Hannu Kankaanranta[13,14,15], Johannes Kettunen [5,6,7], Klaus Bønnelykke [4,16], Aarno Palotie [2,17,18] & Antti Mäkitie [1,3]

The coexistence of asthma and chronic rhinosinusitis with nasal polyposis (CRSwNP) is associated with allergic phenotypes, disease severity and failure of first-line treatment for both asthma and CRSwNP. Recent studies have highlighted shared genetic components for these diseases. To better understand this shared component, we perform genome-wide meta-analyses of asthma (n = 71,481), CRSwNP (n = 9626) and chronic rhinosinusitis without nasal polyposis (CRSsNP, n = 15,448) in FinnGen and UKB (685,602 controls). We detect 131 genomic associations, including 17 novel loci for asthma, 33 novel loci for CRSwNP, and one for CRSsNP. A shared impact on asthma and CRSwNP is observed at 71 loci. A cross-trait meta-analysis using all disorders further implicates 17 loci associated with asthma or asthma and CRSwNP. We also find 17 nonsynonymous associating variants, including a novel *TP63* missense variant association with CRSwNP (OR = 1.519 [1.331–1.734]). Gene set analyses confirm enrichment of genes involved with type 2 inflammation, Jak-STAT signaling, and FOXP3 signaling. Our results highlight new shared and separate genetic pathways for CRSwNP and asthma. These provide several avenues of further investigation in functional and epidemiological follow-up, and evidence for immunological and non-immunological mechanisms behind both diseases.

Chronic rhinosinusitis (CRS) and asthma each affect about 9–12% of the population[1]. CRS is a heterogenous diseases characterized by persistent inflammation of the nasal and sinus passages, with symptoms lasting over 12 weeks[1]. Asthma is a heterogenous disease characterized by inflammation, bronchial hyperreactivity and symptoms. They both have a severe impact on quality of life, morbidity and economy[1]. The overlap between asthma and CRS is ~50%[2]. Co-morbid asthma and CRS is associated with increased morbidity, suffering and costs[3], necessitating studies on the pathogenesis of asthma and CRS to reduce disease burden.

Asthma encompasses various subtypes, with allergic and non-allergic eosinophilic asthma being the most common[4]. CRS has two main phenotypes: with nasal polyps (CRSwNP) and without nasal polyps (CRSsNP)[1]. CRSwNP is characterized by the presence of nasal polyps, edematous non-malignant protrusions of the mucosal membranes lining the middle meatal and ethmoidal region of the sinonasal tract. With an estimated prevalence of 1.9–3.5 % from cross-sectional studies[1,5,6], about half of CRSwNP patients require specialist intervention due to poor disease control and the need for advanced therapy[1,2,7]. CRSwNP is underdiagnosed in the general population, and the true

A full list of affiliations appears at the end of the paper. *A list of authors and their affiliations appears at the end of the paper.
✉e-mail: elmo.saarentaus@helsinki.fi

**Table 1 | Participant counts in case and control groups**

| Phenotype | FinnGen (n) | UKB (n) | Meta-analysis (n) | Age (y), FG | F%, FG | Age (y), UK | F (%), UK |
|---|---|---|---|---|---|---|---|
| Asthma | 42,163 | 29,318 | 71,481 | 65 (48–76) | 61% | 59 (51–64) | 61% |
| CRSwNP | 6255 | 3371 | 9626 | 68 (57–76) | 38% | 59 (52–64) | 33% |
| CRSsNP | 13,534 | 1914 | 15,448 | 62 (49–73) | 66% | 58 (52–63) | 51% |
| Asthma and/or CRS | 55,905 | 33,060 | 88,965 | 64 (49–76) | 60% | 59 (52–64) | 58% |
| Controls | 321,372 | 364,230 | 685,602 | 63 (47–75) | 55% | 58 (51–63) | 54% |

"Asthma and/or CRS" refers to all cases with any mention of the considered diagnoses (Asthma, CRSwNP, or CRSsNP). "Age (y), FG" refers to participants median age in years at death, or the end of the follow-up period, with interquartile range in parentheses, in the FinnGen study; "Age (y), UK" is that of UKBB. "F (%), FG" refers to the percentage of females in the annotated phenotype, in the FinnGen study; "F (%), UK" is that of UKBB.

prevalence is likely higher[1,5]. Underdiagnosis is likely due to a lack of disease awareness and poor nasal endoscopy facilities at the primary care level.

CRS and asthma are classified by inflammatory endotypes. T helper cell 2, i.e., Type 2 (T2) high asthma and CRSwNP are linked to eosinophils and cytokines, interleukin (IL)-4, IL-5, IL-13, while Type 2 low endotype involves Th17 pathways and neutrophilic inflammation[1]. The vast majority of CRSwNP and most asthma cases show Th2-driven inflammation[1]. Despite the increased knowledge of endotypes of CRSwNP and asthma, there is still lacking knowledge of the genome environmental inteactions leading to the onset of co-morbid CRSwNP and asthma[2,8]. Hence, it is important to study the shared genetics of CRSwNP and asthma to enlighten the etiopathology of these diseases.

T2-high asthma and CRSwNP share similarities in their risk factors, such as genetic loci related to type 2 inflammation pathways and to IgE overproduction, allergic rhinitis, recurrent respiratory infections, bacterial colonization, autoimmunity responses and eosinophilia[9]. There are also different risk factors for CRSwNP and type 2-high asthma, which may explain why some individuals get only type 2-high asthma and some only CRSwNP. Shared environmental contributors include epithelium dysbiosis, cigarette smoke and other air pollutants. A shared genetic component between the conditions has been recognized as well[10,11], and the shared heritability is greatly enhanced in CRSwNP[9,12]. Both asthma and CRS have implicated type 2 inflammation pathways of innate immunity, including IL-4, IL-6, IL-13, and the TSLR pathway[1,12–14].

In asthma, over 212 genomic loci have been identified to date, including loci harboring *GSDMB/ORMDL*, *HLA*, *TSLP*, *IL1RL1/Il18R*, and *IL33*[10,15–18], confirming the important role of type 2 inflammatory pathways. For CRSwNP, a genome-wide association study (GWAS) in Icelandic and English populations detected a protective gain-of-function variant in *ALOX15*, suggesting an important role for LOX-high pathways[9]. Recent genetic analyses[9,12] in Finland and the UK have implicated 17 genomic loci associated with CRSwNP. Importantly, beyond observed clinical overlap, 10 of the 17 CRSwNP loci also associate with asthma. Our previous in silico analyses[12] using LD score regression[19] established a genetic correlation of 68.7% between asthma and CRSwNP, highlighting both the similarity and potential heterogeneity of heritable factors in one or both diseases. Several of the shared pathways between asthma and CRSwNP have been targeted in biologic therapeutics[7,20,21], warranting further study of the shared biological aspects of the diseases.

In this work, we show that the shared genetic findings between asthma and nasal polyposis highlight shared and distinct biological processes and structures relevant to both diseases, offering clues to mechanisms of treatment resistance and disease subtypes. To understand the shared genetic contribution, we conduct genome-wide analyses of asthma, CRSwNP, and CRSsNP in FinnGen and the UK Biobank (UKB). Independent analyses of co-morbid disorders have been showed to have a potential to misrepresent shared etiologies[22]. Here, we leverage a previously established approach[12,17,23] using cross-trait analysis of all three endpoints to emphasize shared genetic risk,

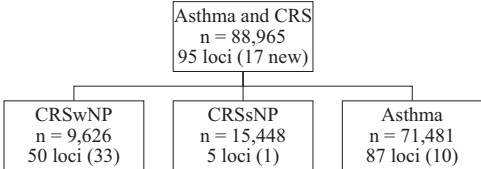

**Fig. 1 | Study overview.** Genome-wide association was carried out on each of the three phenotypes separately, and using an asthma and/or CRS cross-trait phenotype where all participants with asthma or chronic rhinosinusitis (CRS) with nasal polyposis (CRSwNP) or without (CRSsNP) were selected as cases. All four GWASs had the same controls without any of the tested diseases (685,602 controls in total).

instead of limiting analysis to trait-specific associations. A note on the study design is included in the Supplement (Supplementary Note 1).

## Results

### Genome-wide association analysis

We performed genome-wide association studies (GWASs) of asthma, chronic rhinosinusitis with nasal polyposis (CRSwNP), and chronic rhinosinusitis without nasal polyposis (CRSsNP) in FinnGen and UKB (Table 1, Fig. 1). We identified 87 genome-wide significant (GWS) loci associated with asthma (Fig. 2, Supplementary Data 1), of which 10 have not been previously linked with asthma[10,15,17,18,23,24] (Table 2). All novel asthma lead variants had co-directional impact with at least nominal significance ($p < 0.05$) in both cohorts, although heterogeneity for 1p22.1 (harboring *TGFBR3*) was high ($I^2 = 76.3\%$). Of the asthma-associated loci, 26/87 were GWS associated with CRSwNP, including three (5q22.1, 6p21, and 9p24.1, near genes *TSLP*, *HLA*, and *IL33*, respectively) associated with CRSsNP as well. The asthma locus 16q24.3 near *CHMP1A* was also associated with CRSsNP ($OR_{CRSsNP} = 1.068$ [1.043–1.094]). Interestingly, age correction did not appear to have any impact in the FinnGen analysis (Supplementary Fig. 1).

A total of 50 loci associated with CRSwNP (Fig. 2, Supplementary Data 2), including 26 loci also associating with asthma. Most of the CRSwNP loci (33/50) have not been previously reported in the context of CRSwNP (Table 3). Several new loci are relevant to immunological pathways (e.g., 5p13.2 near *IL7R*, 10p15.2 near *IL2RA*, and 22q12.3 near *IL2RB*), including the allergic asthma locus *IL4R* ($OR_{CRSwNP} = 1.151$ [1.104–1.199], $p = 2.54e-11$). Effect heterogeneity between FinnGen and UKB was notable ($I^2 > 50\%$) for ten loci, although co-directional effects were observed in both cohorts for nine of these loci. The 9q22.3 locus did not reach significance in the UKB ($p_{UKB} = 0.17$ for lead variant rs7044307-G near *SEC61B*).

For CRSsNP, we identified five GWS loci (Supplementary Fig. 2, Supplementary Data 3), including one novel association at 9q33.3, near *NEK6*. The locus, previously linked to allergic rhinitis and vitiligo, associated with CRSsNP ($OR = 1.069$ [95 % CI 1.044–1.094], $p = 2.28e-8$). The CRSsNP association for the 5q22.1 locus near *TSLP* exhibited heterogeneity ($I^2 = 68.8\%$) between FinnGen and UKB, and effect

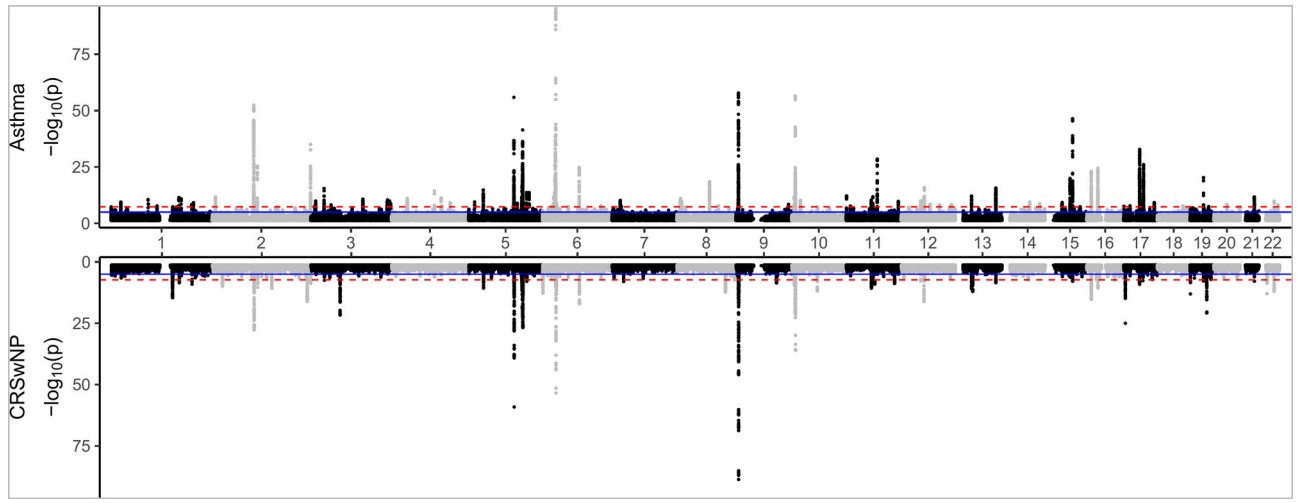

**Fig. 2 | Genome-wide meta-analysis results for 7,268,979 variants in FinnGen and UKB.** Negative log10 *p* values from genome-wide association meta-analysis of asthma (top, 71,481 cases vs 685,602 controls) and CRSwNP (bottom, 9626 cases vs 685,602 controls). The red dashed line denotes genome-wide significance (*p* < 5e-8). *P* values are derived from one-sided $\chi^2$-distribution with one degree of freedom.

**Table 2 | Loci lead variants from genome-wide meta-analysis of asthma in FinnGen and the UK Biobank, not previously implicated with asthma**

| Locus | RSID (EA) | EAF | Gene | Consequence | OR | 95% CI | *p* | *I²* |
|---|---|---|---|---|---|---|---|---|
| 1p22.1 | rs114425738-C | 3.1 | *TGFBR3* | intron | 1.127 | (1.087–1.167) | 2.71E-11 | 76.3% |
| 1p13.2 | rs6679677-A | 14.6 | *RSBN1* | upstream | 1.048 | (1.030–1.065) | 1.85E-08 | 0% |
| 3p25.3 | rs59105104-G | 69.1 | *ATP2B2* | intron | 1.035 | (1.022–1.047) | 1.74E-08 | 0% |
| 5q31.1c | rs5742913-A | 10.1 | *TCF7* | missense | 1.059 | (1.040–1.078) | 2.94E-10 | 0% |
| 8q22.3 | rs646836-T | 35.9 | *NCALD* | intron | 1.034 | (1.022–1.047) | 1.70E-08 | 0% |
| 15q25.3 | rs12916001-G | 72.3 | *AKAP13* | intron | 1.034 | (1.021–1.048) | 4.35E-08 | 0% |
| 17p13.3 | rs7222858-C | 41.7 | *SLC43A2* | intron | 1.035 | (1.022–1.048) | 2.85E-08 | 0% |
| 17p12 | rs62073587-A | 52.2 | *TTC19* | intron | 1.032 | (1.020–1.044) | 1.73E-08 | 2.3% |
| 20q11.21 | rs6119231-A | 14.8 | *KIF3B* | upstream | 1.046 | (1.030–1.062) | 4.74E-09 | 0% |
| 22q13.31 | rs4823863-A | 34.8 | *GRAMD4* | intron | 1.036 | (1.023–1.049) | 6.26E-09 | 0% |

"Locus" denotes the locus ID, based on the chromosome band (see Supplementary Data 5 for enumeration). RSID denotes the variant, with effect allele (EA) marked. EAF: Effect allele frequency (%) in FinnGen R9. Gene: nearest gene to the lead variant. Odds ratios (OR) are derived from mixed logistic regression with confidence interval denoted at α = 0.05 (95% CI). P values (*p*) are derived from one-sided $\chi^2$-distribution with 1 degree of freedom. Heterogeneity between FinnGen and UKB is denoted by the heterogeneity index *I²* (>50% suggesting heterogeneous effect).

estimates were contra-directional for the lead variant rs79411074-G. Conversely, the remaining four loci demonstrated co-directional and homogeneous effects between the cohorts.

**Cross-trait analysis**

Given the previously observed[9,12] shared loci and genetic correlation between asthma and CRS, we conducted a further GWAS with all participants with asthma, CRSwNP and CRSsNP as cases (*n* = 88,965). This enhances the analysis of shared genetic risk, at the expense of single phenotype-specific genetic risk, as seen in previous studies involving asthma and allergic diseases[17,23]. The asthma and/or CRS GWASs were run in FinnGen and UKB separately with the same pipelines as above, and the same controls. The GWAS resulted in 95 GWS loci (Supplementary Data 4), of which 17 were not detected in single-phenotype analyses (Table 4).

In our genome-wide analyses, we observed inflation of test statistics, especially concerning asthma and the cross-trait phenotype, in both the meta-analyses and the original GWASs carried out in FinnGen and UKB as indicated genomic lambda values exceeding 1.1 (Supplementary Table 1). However, as the LD Score Regression intercept values were close to 1, the inflation is likely due to a polygenic signal[19,25]. Also, the Q-Q plots exhibited a distribution of observed *p* values

commonly observed in genome-wide analyses with significant hits, suggesting no substantial deviation from the expected distribution (Supplementary Figs. 1–3). Together with trait-specific analyses, the cross-trait analysis allowed us to identify 131 genomic loci associating with asthma, CRS, or their combination (Supplementary Data 5).

**Bayesian phenotype grouping**

To further elucidate the effects of shared genetic variants on specific phenotypes, we implemented a Bayesian phenotype grouping approach employing MetABF[26]. This method utilizes a Bayesian tree analysis framework to identify the most probable underlying model from a given set while accounting for overlapping cases, in addition to prevalence differences. The models considered were: SHARED, with an equal or highly similar true effect on all tested phenotypes; phenotype-specific models (e.g., CRSwNP), with a true non-zero effect in only one phenotype; and NULL, where there is no true effect for the tested phenotypes.

Through this analysis, we identified that 13 of the 17 loci, which did not appear in single-phenotype GWAS, contributed to both asthma and CRSwNP. Of the remaining 4 loci, three appeared to affect only asthma: 2q24.1 near *GPD2*, 8p23.1a near *SGK223*, and 8p23.1d near *DEFB136* (Fig. 3). Interestingly, while lacking CRSwNP impact, these

**Table 3 | Loci lead variants from genome-wide meta-analysis of CRSwNP in FinnGen and the UK Biobank, not previously reported with nasal polyps**

| Band | RSID (EA) | EAF | Gene | Consequence | OR | 95% CI | p | I² |
|---|---|---|---|---|---|---|---|---|
| [a]1q24.2 | rs1773561-T | 26.3 | CD247 | intron | 1.100 | (1.066–1.136) | 3.19E-09 | 0% |
| [a]1q32.1a | rs113752715-G | 3.8 | PTPRC | intron | 1.242 | (1.159–1.330) | 7.80E-10 | 0% |
| 2p23.3 | rs11894248-A | 70.7 | NCOA1 | intron | 1.109 | (1.074–1.144) | 1.18E-10 | 0% |
| [a]2q13 | rs72836346-G | 90.4 | BCL2L11 | intron | 1.180 | (1.123–1.239) | 4.20E-11 | 0% |
| 2q37.1 | rs6704768-A | 52.2 | GIGYF2 | intron | 1.129 | (1.097–1.163) | 6.58E-17 | 93% |
| [a]2q37.3 | rs113372064-T | 7.9 | D2HGDH | 3' UTR | 1.182 | (1.120–1.249) | 1.58E-09 | 0% |
| [a]3p22.3 | rs9828592-T | 50.6 | GLB1 | intron | 1.084 | (1.053–1.116) | 3.24E-08 | 0% |
| [a]5p13.2 | rs10075764-G | 39.5 | IL7R | regulatory | 1.107 | (1.075–1.141) | 2.06E-11 | 0% |
| [a]5q31.3 | rs1821263-C | 47.7 | NDFIP1 | intergenic | 1.087 | (1.056–1.120) | 1.14E-08 | 0% |
| [a]6p25.3b | rs1050979-A | 52.5 | IRF4 | 3' UTR | 1.113 | (1.082–1.146) | 2.01E-13 | 58.3% |
| [a]6q15 | rs62408224-G | 25.0 | BACH2 | intron | 1.152 | (1.115–1.191) | 1.02E-17 | 76.2% |
| 7p15.1 | rs12531540-T | 48.9 | JAZF1 | intron | 1.087 | (1.056–1.119) | 9.64E-09 | 12% |
| 8q24.12 | rs4618726-A | 55.7 | COLEC10 | intron | 1.111 | (1.080–1.144) | 4.85E-13 | 0% |
| 8q24.3 | rs116877855-C | 95.5 | PTP4A3 | intron | 1.196 | (1.122–1.275) | 4.42E-08 | 20.5% |
| 9q22.33 | rs7044307-A | 27.5 | SEC61B | intron | 1.111 | (1.073–1.150) | 2.65E-09 | 75.1% |
| [a]10p15.1 | rs4749920-T | 67.4 | IL2RA | intron | 1.105 | (1.071–1.140) | 2.71E-10 | 31.3% |
| 10q21.2 | rs10995258-T | 35.4 | ZNF365 | intron | 1.115 | (1.082–1.150) | 1.47E-12 | 0% |
| 11q13.3 | rs12279397-A | 49.3 | ANO1 | intron | 1.095 | (1.063–1.127) | 7.56E-10 | 0% |
| 11q23.3 | rs12365699-G | 84.3 | CXCR5 | intergenic | 1.124 | (1.082–1.167) | 1.51E-09 | 0% |
| [a]12q13.11 | rs11168249-C | 50.9 | HDAC7 | intron | 1.086 | (1.055–1.118) | 1.82E-08 | 60% |
| [a]13q14.11a | rs9594366-T | 34.2 | COG6 | downstream | 1.111 | (1.077–1.146) | 1.17E-11 | 0% |
| 13q14.11b | rs7327510-G | 38.2 | AKAP11 | intron | 1.113 | (1.081–1.147) | 9.01E-13 | 0% |
| [a]13q32.3 | rs2182885-G | 44.7 | UBAC2 | intron | 1.092 | (1.060–1.124) | 3.26E-09 | 11.1% |
| 14q23.3 | rs6573559-G | 69.2 | MTHFD1, ZBTB25 | intron | 1.089 | (1.056–1.124) | 4.02E-08 | 0% |
| [a]15q22.33 | rs17293632-C | 73.8 | SMAD3 | intron | 1.099 | (1.063–1.135) | 1.21E-08 | 36.2% |
| [a]16p12.1 | rs2891058-G | 15.8 | IL4R | intron | 1.151 | (1.104–1.199) | 2.54E-11 | 0% |
| 16q22.1 | rs1728797-A | 18.4 | ZFP90 | intron | 1.108 | (1.068–1.149) | 4.07E-08 | 0% |
| 17q21.32 | rs41444548-C | 93.1 | TBX21 | intron | 1.164 | (1.104–1.228) | 2.20E-08 | 66.5% |
| 19p13.3 | rs4807542-A | 23.4 | GPX4 | synonymous | 1.149 | (1.108–1.192) | 8.75E-14 | 69.1% |
| 19q13.11 | rs11671925-A | 23.8 | SLC7A10 | upstream | 1.109 | (1.070–1.150) | 1.68E-08 | 0% |
| 21q22.12 | rs8129030-T | 34.7 | RUNX1 | intron | 1.092 | (1.059–1.125) | 1.31E-08 | 20.6% |
| 22q11.21 | rs1978060-A | 35.9 | TBX1 | intron | 1.122 | (1.088–1.157) | 1.15E-13 | 44.4% |
| [a]22q12.3 | rs2543537-T | 51.0 | IL2RB | intron | 1.111 | (1.079–1.144) | 9.53E-13 | 0% |

RSID denotes the variant, with effect allele (EA) marked. EAF: Effect allele frequency (%) in FinnGen R9. Gene: nearest gene to the lead variant. Odds ratios (OR) are derived from mixed logistic regression with confidence interval denoted at $\alpha = 0.05$ (95% CI; ±1.96*SE). P values (p) are derived from one-sided $\chi^2$-distribution with 1 degree of freedom. Heterogeneity between FinnGen and UKB is denoted by the heterogeneity index $I^2$ (with $I^2 > 50\%$ suggesting heterogeneous effect).
[a]GWS associated with asthma in FinnGen-UKB meta-analysis.

three loci showed possible shared effect to CRSsNP (Supplementary Fig. 2), and 2q24.1 in particular. The impact of the 3p14.3 locus near *PPA1P16* could not be definitively categorized as either shared or asthma-specific. Among the shared impact loci was 7q22.3, lead by the missense variant rs6967330-G in the viral receptor-encoding *CDHR3*; this variant has been previously linked to early childhood asthma with severe exacerbations[27] and CRS[28].

The Bayesian analysis showed an even greater shared signal between asthma and CRSwNP than initial results revealed. Specifically, the cross-trait asthma and/or CRS GWAS implicated 43 asthma loci that did not appear in the CRSwNP GWAS. Further examination of the phenotype-specific impact of these loci indicated that 26 loci likely exert a shared effect on both asthma and CRSwNP (Fig. 4). Intriguingly, five asthma loci showed a significantly higher effect on CRSwNP: 6q24.2 near *PHACTR2* ($OR_{CRSwNP} = 1.14$ [1.07–1.21]), 5q31.1 near *TCF7* ($OR_{CRSwNP} = 1.11$ [1.05–1.17]), 11q24.3 near *ETS1* ($OR_{CRSwNP} = 1.12$ [1.07–1.17]), 12q13.3 near *STAT6* ($OR_{CRSwNP} = 1.09$ [1.04–1.13]), and 17p12 near *TTC19* ($OR_{CRSwNP} = 1.07$ [1.04–1.11]). Considering CRSsNP as well (Supplementary Fig. 3), a probable asthma-specific effect

(PP > 50 % for ASTHMA model) was apparent for only 6 of 43 loci: 4p14, 4q24, 7p21.1, 8p23.1b, 8p23.1c, and 10p12.31.

To analyze trends of shared and distinct impact on asthma and CRSwNP, we employed linemodels[29] for probabilistic clustering of variants based on estimated effect sizes. This method corrects for correlation between cases of different phenotypes specified for the FinnGen and UKB subpopulations. We assessed three grouping models: "ASTHMA," suggesting a non-zero effect on asthma only; "CRSwNP," indicating a non-zero effect exclusively on CRSwNP; and "BOTH," which posits a highly correlated and proportional effect on both conditions.

Overall, we identified a total of 131 genomic loci (Supplementary Data 5), of which 71 exhibited proportional associations with both asthma and CRSwNP, 14 were specific to CRSwNP, 20 were unique to asthma, and an additional 26 remained unclassified (Fig. 5). Notably, the unclassified loci displayed co-directional impacts and were presumed to associate with asthma, with some potentially influencing CRSwNP as well. Of the unclassified loci, 1p22.1, 15q22.2, and 17q21.33 were classified as shared in the MetABF analysis (Fig. 4). Importantly,

**Table 4 | Lead variants for 17 loci detected uniquely in asthma and/or CRS genome-wide association meta-analysis**

| RSID (EA) | EAF | Gene | Cross-trait OR (CI) | p | I² | Asthma OR (CI) | CRSwNP OR (CI) |
|---|---|---|---|---|---|---|---|
| rs62176646-C | 65.1 | GPD2 | 1.031 | 1.59E-08 | 10.3% | 1.032 (1.019-1.044) | 1.001 (0.970-1.033) |
| rs4664446-G | 43.5 | DPP4 | 1.030 | 2.94E-08 | 0.0% | 1.031 (1.012-1.050) | 1.036 (1.005-1.067) |
| rs62184059-C | 95.6 | CTLA4 | 1.077 | 5.65E-10 | 0.0% | 1.071 (1.044-1.100) | 1.160 (1.086-1.239) |
| rs6796752-C | 29.9 | SLMAP | 1.032 | 1.91E-08 | 21.1% | 1.033 (1.020-1.045) | 1.023 (0.991-1.057) |
| rs10936753-G | 74.0 | CEP97 | 1.032 | 3.65E-08 | 0.0% | 1.033 (1.020-1.046) | 1.039 (1.006-1.074) |
| rs16853094-G | 68.6 | MECOM | 1.035 | 1.88E-09 | 0.0% | 1.033 (1.020-1.046) | 1.050 (1.017-1.085) |
| rs61665417-T | 25.8 | PTTG1 | 1.032 | 3.75E-08 | 0.0% | 1.032 (1.019-1.046) | 1.079 (1.044-1.116) |
| rs6965423-T | 50.0 | PCLO | 1.030 | 1.15E-08 | 0.0% | 1.027 (1.016-1.039) | 1.035 (1.005-1.065) |
| rs6967330-G | 73.2 | CDHR3 | 1.036 | 9.71E-09 | 0.0% | 1.029 (1.015-1.043) | 1.045 (1.009-1.081) |
| rs2921057-T | 46.9 | SGK223 | 1.032 | 1.04E-09 | 17.0% | 1.036 (1.024-1.048) | 1.010 (0.981-1.040) |
| rs28510449-C | 37.0 | DEFB136 | 1.032 | 2.57E-09 | 0.0% | 1.038 (1.025-1.050) | 0.998 (0.968-1.027) |
| rs10984766-A | 57.8 | CDK5RAP2 | 1.030 | 1.79E-08 | 64.9% | 1.029 (1.017-1.041) | 1.030 (1.000-1.061) |
| rs2901610-T | 50.2 | EXOC6 | 1.031 | 1.47E-09 | 46.7% | 1.030 (1.018-1.041) | 1.060 (1.030-1.091) |
| rs12716974-G | 50.2 | KCTD13 | 1.030 | 8.06E-09 | 91.3% | 1.029 (1.017-1.040) | 1.049 (1.019-1.080) |
| rs7184567-C | 60.7 | STX1B | 1.034 | 9.36E-11 | 0.0% | 1.029 (1.017-1.041) | 1.074 (1.043-1.107) |
| rs11645550-C | 51.5 | RP11-457D20.2 | 1.029 | 2.99E-08 | 0.0% | 1.024 (1.013-1.036) | 1.036 (1.006-1.067) |
| rs8099412-T | 55.9 | CD226 | 1.028 | 4.22E-08 | 0.0% | 1.030 (1.019-1.042) | 1.067 (1.037-1.099) |

Cross-trait OR (CI): odds ratio and 95% confidence interval for variant on the GWAS of asthma and/or CRS. RSID denotes the variant, with effect allele (EA) marked. EAF: Effect allele frequency (%) in FinnGen R9. Gene: nearest gene to the lead variant. P: lead variant p-value in asthma and/or CRS GWAS. P values (p) are derived from one-sided χ²-distribution with 1 degree of freedom. I²: Heterogeneity index between cross-trait analysis in FinnGen and that of UKB (I² > 50% suggesting heterogeneous effect). Asthma OR (CI): variant odds ratio of asthma, with 95% confidence interval (±1.96*SE). CRSwNP OR (CI): variant odds ratio of CRSwNP, with 95% confidence interval (±1.96*SE).

among loci shared by both conditions, the effect on CRSwNP was consistently stronger than that on asthma, with a slope of 1.88 in log-odds space in favor of CRSwNP, even if 95% confidence intervals of individual loci still overlapped with that of asthma. This tendency for a more pronounced effect on CRSwNP was also observed in previously asthma-specific loci, such as 16p13.3 near *CLEC16A* and 16p12.1 near *IL4R*. Such findings align with prior observations[9] regarding loci like 9p24.1 near *IL33*, 5q22.1 near *TSLP*, and 10p14 near *GATA3*, which were among the first identified as related to CRSwNP[17,18,23]. In our disease-specific GWASs, 22.9% (20/87) of asthma loci were determined not to impact CRSwNP. Similarly, 28% (14/50) of CRSwNP loci exhibited no significant effect on asthma despite the significantly lower case count of CRSwNP, further emphasizing the nuanced genetic landscape shared between these conditions.

### Fine-mapping of potential causal variant

We performed fine-mapping of non-*HLA* loci in the FinnGen cohort using SuSIE[30]. This fine-mapping identified high-quality credible sets for 46 genomic loci (Supplementary Data 6–9). Among credible sets, we selected predicted non-synonymous variants, and tested for their specific impacts in the UKB if they had been omitted in the original analysis due to rarity. These results were meta-analyzed, with GWS variants reported in Table 5. This resulted in known high-impact variants, such as rs144651842-A in *IL4R*, rs2305479-T in *GSDMB*, and rs113135335-G in *BCL2L11* for asthma, and rs34210653-A in *ALOX15* for CRSwNP, and novel variant associations such as the atopic dermatitis-associated[31] rs183884396-A in *TESPA1* for asthma.

Fine-mapping singled a low frequency (AF = 1.39%) *TP63* variant rs190865056-A that significantly increases risk for CRSwNP (OR = 1.52 [95% CI 1.331–1.734]), with no significant impact on asthma. This variant is reported to be Finnish-enriched in the gnomAD database[32], suggesting ancestry overlap for UKB participants with the variant. In most of the 13 *TP63* transcripts, rs190865056-A is predicted to result in an isoleucine-asparagine missense mutation. A phenome-wide association performed for this variant in the FinnGen R9 PheWEB (n = 2272 phenotypes) also detected a co-directional impact on basal cell carcinoma (OR = 1.28 [95% CI 1.17–1.40], p = 5.1e-08). The association with basalioma was investigated in all 130 non-*HLA* loci, where it was further associated with six loci (1p13.2, 1q21.3, 3p13, 6q15, 12q13.2, and 16q24.3), five of which had opposite direction of effect.

### Colocalization analyses

In order to study the differences between loci groupings according to effect on asthma and CRSwNP, we also tested the impact of non-*HLA* loci credible sets from FinnGen. These were associated and colocalized with FinnGen R9 endpoints selected based on previously observed[17,18,23] asthma comorbidities: atopic dermatitis, allergic endpoints, and autoimmune diseases (Supplementary Data 10). Atopic dermatitis had probable colocalization (posterior probability > 0.9) with 7 loci, of which 6 had a shared asthma/CRS impact, and one unassigned. Allergic phenotypes colocalized with 16 loci, mostly having a shared asthma/CRS impact, with the exception of 3p21.3 locus harboring *EEFSEC*, which our previous analysis labeled as asthma-specific. Autoimmune diseases colocalized with five loci (5p13.2, 11q13.5, 12q13.2, 12q13.3, and 17q21.32), all with shared asthma/CRS impact. For three of these five loci (5p13.2, 12q13.3, and 17q21.32) the nearest genes (*IL7R*, *STAT6*, and *TBX21*, respectively) have previously reported Mendelian immune deficiency variants. In total, 10 loci harbored genes with such reported Mendelian immune deficiency variants[33] (Supplementary Data 5). These loci with immune deficiency-linked genes shared impact on both asthma and CRS in 9 of 10 loci and one, 10p15.1, remained unassigned, with GWS impact to asthma (OR$_{asthma}$ = 0.804 [95% CI 0.756–0.855], p = 1.90E-12).

We also investigated asthma subtype association among asthma-associated loci (Supplementary Data 10). Here, we

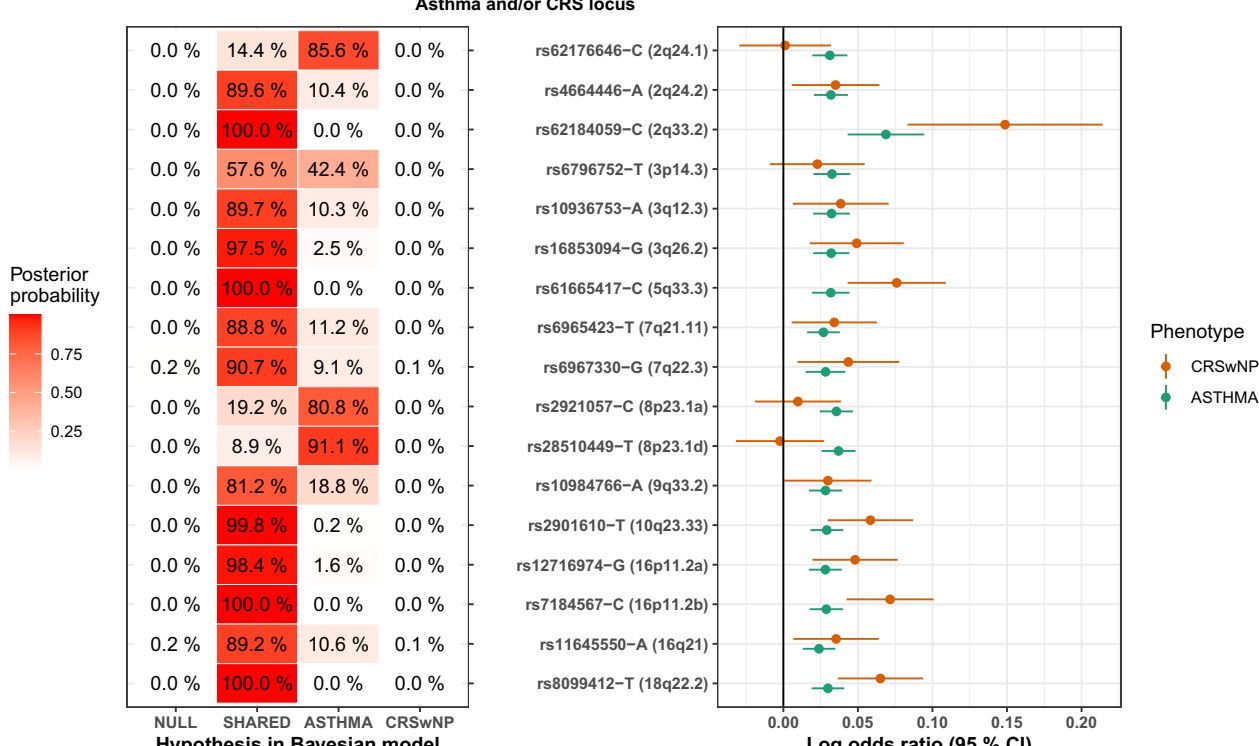

**Fig. 3 | Bayesian cross-trait analysis of lead variants of loci not detected in asthma or CRS analyses (Table 4; 88,965 cases vs 685,602 controls).** Left panel: Bayesian analysis indicating the most probable model. NULL: Null model. SHARED: the effect is identical or very similar for both asthma and CRSwNP. ASTHMA: there is an effect on asthma but no effect on CRSwNP. CRSwNP: there is an effect on CRSwNP but no effect on asthma. Right panel: Forest plot, presenting natural logarithm of odds ratio, with 95 % confidence intervals (±1.96 standard errors), of lead variants in phenotype-specific analyses.

colocalized asthma-associated loci with subtype analyses from FinnGen R9 PheWEB. Specifically, we investigated colocalization with childhood asthma, defined as onset before the age of 16 ($n$ = 5865), and asthma-related infections ($n$ = 16,018). Childhood onset was colocalized (PP > 0.9) with 10 loci, all with shared impact to CRS, and 8 colocalized with allergic diseases. Asthma-related infections were colocalized (PP > 0.9) with 10 shared loci and one unassigned (17q21.33). Interestingly, loci harboring genes with Mendelian immune-deficiency variants did not colocalize with asthma-related general infections at all, suggesting the existence of genetically distinct pathways.

Thirdly, we colocalized (PP > 0.9) credible sets with eQTLs identified in gene expression data in 44 non-gonadal tissues from GTEx[34] (Supplementary Data 11). 16 loci colocalized with differing expression of 29 genes among all 44 tissues. In lung tissue samples, three loci colocalized with gene expression of three genes: 4q24 (near *NPNT*) had codirectional effect with expression of long non-coding RNA (lncRNA) ENSG00000250740; 12q13.2 (near *RPS26*) had opposite effect on *SUOX* and *RPS26* expression; and 17q21.1 (near *GSDMB*) had codirectional effect with *MED24* expression.

**In silico analyses**

In order to further study the shared and distinct genes and pathways, we performed Multi-marker Analysis of GenoMic Annotation (MAGMA, v1.10)[35] to identify genes and gene sets enriched in each data set. This analysis summarizes GWAS results over gene regions to identify associated genes rather than single alleles. Genes were downloaded from NCBI RefSeq ($n$ = 19526). For gene set analysis, we used canonical gene sets ($n$ = 7561) described at MSigDB 7.0[36]. MAGMA highlighted a total of 330 genes, of which 121 genes associated with CRSwNP, 7 associated with CRSsNP, and 246 associated with asthma (Supplementary Data 12). MAGMA identified 246 genes in the asthma

and/or CRS cross-trait analysis, of which 34 were not significant in single-phenotype MAGMA analyses. The gene set analysis identified 49 sets of genes with increased association to phenotypes (Supplementary Data 13), including known asthma-associated pathways such as Jak-STAT signaling[37], innate lymphoid cell lines[38], and *RUNX3* signaling[39]. CRSwNP was also associated with Jak-STAT and RUNX3 signaling, in addition to the *NFAT* transcription factor pathway and others. A notable shared signal was also confirmed with genes associated with atopic dermatitis mechanisms and therapies, a corollary to clinical manifestations[40–43].

Immunologically relevant tissues were enriched with asthma (Supplementary Table 2). We performed a tissue-specific enrichment analysis using MAGMA, associating tissue-specific gene expression profiles from 54 tissues from GTEx v8[34] with disease-gene associations. This linked asthma-associated genes with spleen, whole blood, and EBV-transformed lymphocyte tissues, and spleen and whole blood gene expression with asthma and/or CRS. No tissue was significantly linked with CRSwNP or CRSsNP after multiple testing correction. Interestingly, lung tissue gene expression was less associated with asthma-associated genes than lymphoid cell lines, such as blood and spleen, and lung tissue association was not significant after multiple testing correction. This analysis is limited due to underrepresentation of airway tissues.

Finally, we performed LD score regression to establish estimates of heritability (Supplementary Table 1) and genetic correlation between the phenotypes (Supplementary Table 3). Liability scale SNP-based heritability for asthma was estimated at 17.1 % in UKB and 13.7 % in FinnGen. For CRSwNP we estimate a higher SNP-based heritability of 24.1 % in UKB and 33.2 % in FinnGen, which is also evidenced by similar numbers of genetic loci with relatively few CRSwNP cases. In FinnGen CRSsNP had an estimated SNP-based heritability of 6.6 %. In UKB, CRSsNP heritability was not significantly different from 0 and was

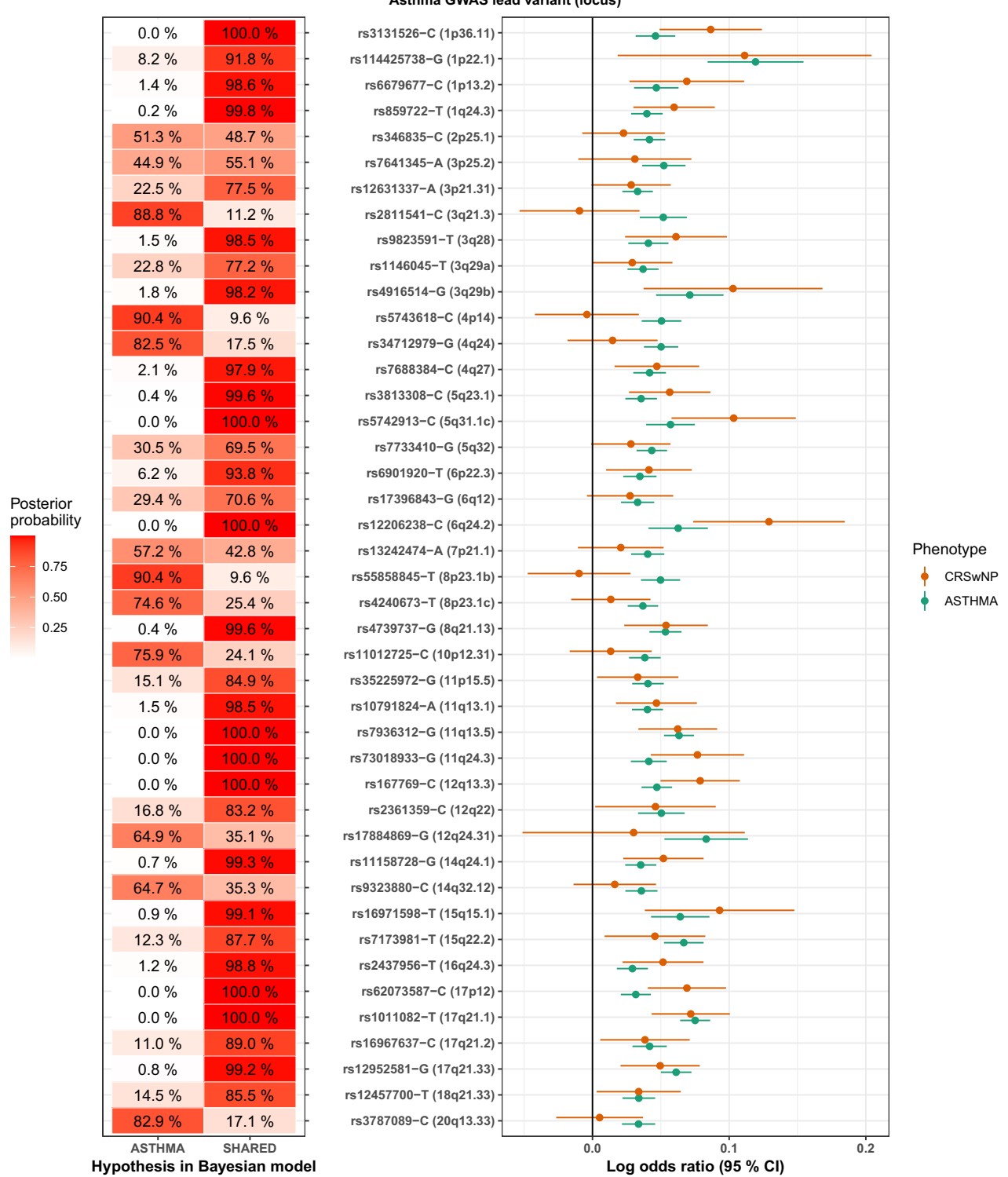

**Fig. 4 | Shared effects in Bayesian cross-trait analysis of lead variants seen in the FinnGen-UKB asthma GWAS (71,481 cases vs 685,602 controls) but not CRSwNP GWAS (9626 cases vs 685,602 controls).** Left panel: Bayesian analysis indicating the most probable model. SHARED: the effect is identical or very similar in both asthma and CRSwNP. ASTHMA: there is an effect on asthma but no effect on CRSwNP. Right panel: Forest plot of natural logarithm of odds ratios, with 95 % confidence intervals (±1.96 standard errors), of lead variants in phenotype-specific analyses.

excluded from genetic correlation analyses. Genetic correlation between FinnGen and UKB analyses has a confidence interval upper bound of 100% for CRSwNP ($p = 1.66E-29$), whereas for asthma the correlation between cohorts was high but not complete ($r_G = 86.5\%$ [95 % CI 80.6–92.3 %], $p = 7.04E-185$). Genetic correlation between asthma and CRSwNP was lower in UKB (61.4 %) and FinnGen (52.72 %)

than our previous estimate (68.7 %)[12]. In our previous study, SAIGE was used for genome-wide association, which is based on saddlepoint approximation and has been reported to inflate estimated effects. This has potentially also inflated our previous genetic correlation estimate, and we hold our present estimates based on REGENIE genomic analysis to be more reliable.

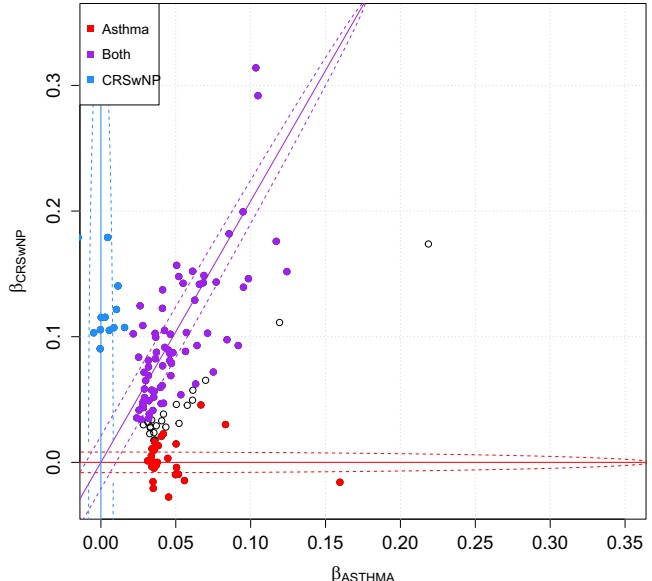

**Fig. 5 | Genomic loci lead variants clustering impacts on asthma and CRSwNP.** Scatterplot of log-odds effect on asthma in FinnGen-UKB meta-analysis ($\beta_{ASTHMA}$) on the horizontal axis, and log-odds effect on CRSwNP in FinnGen-UKB meta-analysis ($\beta_{CRSwNP}$) on the vertical axis. The red line denotes the "Asthma only" model, where the true log-odds effect is non-zero only for asthma. Similarly, the blue line denotes the 'CRSwNP only' model, with the variant's true effect is assumed to only exist for CRSwNP. Purple denotes the model for a shared effect ("Both"), where a non-zero effect to both asthma and CRSwNP is assumed to exist and fit or correlate with a fitted slope ($k = 1.880129$) passing through the origin. All three models have an assumed correlation of 0.999, with dashed lines representing expected deviation with the correlation. Each circle represents a locus lead variant, with coloring according to the model where the linemodels algorithm predicts the variant to belong with ≥90% posterior probability. Variants that do not meet this criterion in any model are represented by circles filled with white and outlined in black.

## Discussion

Asthma and CRS are common diseases with often lifelong burden and about half of CRS cases have asthma[44], possibly due to shared pathomechanisms[45,46]. Recent advances in biomarker research, driven by biological therapies, have identified various CRS clusters with distinct inflammatory profiles, predominantly Th2 high IL-5 levels are associated with a nasal polyp phenotype and increased asthma prevalence[47]. Studies show that Th2-driven CRSwNP is present in about 80% of European patients, whereas Th1/Th17 inflammation is more common in Asian patients[45]. Asthma with nasal polyps is more challenging to manage and prone to frequent exacerbations[45]. Poor symptom control in severe asthma patients correlates with high sinus CT scores[48]. Biological treatments targeting type 2 immune responses have been effective in improving control of both asthma and CRS and lowering the necessity for further sinus surgeries[45].

Here we show for the first time a systematic analysis of the shared genetic contribution between asthma and CRSwNP to elucidate the observed clinical overlap of the conditions. We report the first genome-wide meta-analyses of asthma and CRSwNP in the Finnish and British populations. We detect 17 new loci associating with asthma, 33 new loci associating with CRSwNP, and one locus associating with CRSsNP. Comparing effect estimates allows us to distinguish a total of 131 loci, of which 20 associate only with asthma, 14 only with CRSwNP, and 71 loci with both. We also provide evidence of a trend of shared loci having a higher absolute risk increase for CRSwNP, including many loci traditionally considered asthma-specific, such as 5q22.1 near *TSLP*, or 9p24.1 near *IL33*. This is in line with clinical observations, where some biologics targeting asthma pathways have shown impact on CRSwNP as well[7,21], and even improved impact in patients with both

diseases[20]. The apparent trend of higher genetic CRSwNP impact may stem at least in part from an expected heterogeneity of asthma diagnoses.

Our results indicate that there exist both shared and distinct genetic effects to asthma and CRSwNP, and that these effects can, in most cases, be clearly separated. Previous studies have shown largely similar genetic signals for subtypes of asthma. Here, we present evidence of genetically distinct signals impacting CRSwNP, suggesting separate pathways and mechanisms yet to be identified. CRSwNP-specific loci differ from asthma loci by a lack of association with allergic diseases. By contrast, allergic diseases and immune deficiency phenotypes seemed to largely correlate with impacting both asthma and CRSwNP. Efforts to recognize what constitutes an asthma-specific genetic effect yielded few consistent clues in biological and clinical comorbidities. The apparently higher CRSwNP impact among shared variants may well be explained by more special care-based diagnoses, in addition to a higher heterogeneity of asthma etiologies. Further work is needed to clarify whether these signals can be distinguished downstream as well, which might imply the existence of endophenotypes in both CRSwNP and asthma.

Several loci previously associated with childhood asthma[16,18,49] (e.g., 15q22.33, 16p12.1, and 3q21.3 near genes *SMAD3*, *IL4R*, *EEFSEC*, respectively) were also linked here with a more generally selected asthma. This observation sheds lights on the high but not entirely complete genetic correlation between childhood and adult-onset asthma[18]. Less clear is the mechanism behind childhood asthma loci such as 6p25.3 near *IRF4* and 17q21.32 near *TBX21*, which we associate here with CRSwNP but not asthma. Interestingly, despite CRSwNP usually manifesting in adulthood, some loci associated with childhood asthma also overlap with CRSwNP. This observation emphasizes the complexities of heritable etiologies shared between CRSwNP and both asthma types. The apparently higher CRSwNP impact among shared variants may well be explained by more special care-based diagnoses, in addition to a higher heterogeneity of asthma etiologies. Additionally, while the shared genetic contributions of atopic dermatitis and asthma have previously been well characterized[12,17,23,43], we found no overlap between CRSwNP-specific loci and atopic dermatitis. While epidemiologically expected[50], this absence raises the question of the underlying mechanisms driving the impact of atopic dermatitis, asthma, and CRSwNP, suggesting a common immunological contribution related to type 2 inflammation., as further evidenced by type 2 targeting therapeutics impacting all three diseases[7,21,40]. However, the unique influences of asthma-specific loci remain a matter for exploration, potentially relating to interaction of environmental risk factors – such as obesity[51] and smoking[52].

Among variants with expected direct impact on the polypeptide chain, we identify a rare, Finnish-enriched missense variant rs190865056-A in *TP63* to associate with CRSwNP. Similar to the *ALOX15* missense variant rs34210653-A, rs190865056-A has little apparent impact on the risk for asthma[9]. A member of the p53 family, the p63 protein encoded by *TP63* activates bullous pemphigoid antigen 1 (BPAG-1)[53] and acts as a cell fate determinant of development of limb and epithelium[54,55]. Several high-impact variants have been linked with syndromes impacting limb development and skin conditions such as ectodermal dysplasias[56]. Li and colleagues[57] performed immunohistochemistry and quantitative PCR for nasal polyp tissue of 65 CRSwNP patients and inferior turbinate samples of 19 controls, detecting that p63 expression is elevated in patients with CRSwNP, relates to epithelial hyperplasia in nasal polyp tissue, and is reduced in CRSwNP patients after oral glucocorticoid treatment. While not currently considered oncogenic, p63 is elevated in basaliomas, squamous cell carcinomas, thymomas, urothelial carcinomas, and salivary gland neoplasias[58]. For CRSwNP, the shared impact on basal cell carcinoma of the *TP63* variant appears to be more of an exception, as most CRSwNP loci with an impact on basaliomas had an opposite direction of effect.

**Table 5 | Disease-associated variants predicted to alter the amino acid sequence**

| Phenotype | RSID | Gene | EAF | OR | CI (95%) | P value |
|---|---|---|---|---|---|---|
| ASTHMA | [a]rs61816761-A | FLG | 0.29% | 1.17 | (1.107–1.243) | 4.58E-08 |
| ASTHMA | rs113135335-G | BCL2L11 | 10.8% | 1.10 | (1.072–1.123) | 2.15E-25 |
| ASTHMA | [a]rs5743618-A | TLR1 | 16.0% | 0.95 | (0.937–0.965) | 9.44E-12 |
| ASTHMA | [a]rs16903574-G | FAM105A | 13.0% | 1.06 | (1.040–1.079) | 3.12E-10 |
| ASTHMA | rs183884396-A | TESPA1 | 0.61% | 1.30 | (1.191–1.414) | 9.82E-09 |
| ASTHMA | rs144651842-A | IL4R | 7.79% | 0.882 | (0.857–0.907) | 6.63E-19 |
| ASTHMA | rs11557467-T | ZPBP2 | 44.00% | 0.932 | (0.918–0.946) | 7.53E-38 |
| ASTHMA | rs2305479-T | GSDMB | 45.16% | 0.931 | (0.918–0.945) | 5.40E-39 |
| CRSwNP | rs113135335-G | BCL2L11 | 10.8% | 1.17 | (1.103–1.233) | 1.74E-09 |
| CRSwNP | rs190865056-A | TP63 | 1.39% | 1.52 | (1.331–1.734) | 5.05E-10 |
| CRSwNP | rs6897932-T | IL7R | 33.2% | 0.91 | (0.873–0.943) | 6.32E-09 |
| CRSwNP | rs1050152-T | SLC22A4 | 31.8% | 0.84 | (0.805–0.871) | 3.08E-26 |
| CRSwNP | rs34210653-A | ALOX15 | 0.7% | 0.37 | (0.274–0.489) | 9.44E-26 |
| CRSsNP | rs3184504-C | SH2B3 | 40.85% | 0.93 | (0.909–0.956) | 4.06E-08 |
| CRSsNP | rs602662-A | FUT2 | 41.43% | 1.07 | (1.046–1.100) | 2.92E-08 |
| CROSSTRAIT | rs113135335-G | BCL2L11 | 10.8% | 1.09 | (1.063–1.109) | 2.36E-24 |
| CROSSTRAIT | rs5742913-A | TCF7 | 10.1% | 1.06 | (1.040–1.086) | 5.23E-14 |
| CROSSTRAIT | [a]rs6967330-G | CDHR3 | 27.0% | 1.04 | (1.023–1.049) | 9.71E-09 |
| CROSSTRAIT | [a]rs17884869-A | PITPNM2 | 3.6% | 1.09 | (1.054–1.114) | 1.29E-08 |
| CROSSTRAIT | rs144651842-A | IL4R | 7.77% | 0.89 | (0.870–0.915) | 6.69E-20 |
| CROSSTRAIT | rs11557467-T | ZPBP2 | 44.0% | 0.93 | (0.921–0.946) | 3.95E-43 |

All variants are predicted missense variants except rs61816761-A (stop gain in *FLG*). Phenotype: phenotype of original GWAS, where "CROSSTRAIT" refers to a GWAS with any cases of asthma, CRSwNP or CRSsNP against controls unaffected by these disorders. RSID: Variant RSID and alternate allele. Gene: gene symbol. EAF: Allele frequency in FinnGen (release 9). OR: Odds ratio estimate in a logistic regression model for the denoted phenotype. CI (95%): Confidence interval of OR with alpha = 0.05 (±1.96*SE). *P* value = *P* value from meta-analysis of FinnGen and UK Biobank GWASs of the denoted phenotype, calculated using one-sided $\chi^2$ testing (one degree of freedom) from a t-statistic under a normal approximation.
[a]Lead variant in locus with no good quality credible set.

Limitations of this biobank-scale registry-based study include collider bias (increased chance of diagnosis in the hospital setting) and a smaller proportion of pediatric patients than in the general population. The approach of combining cases and using all three disorders to select controls emphasizes the shared biology, and it is therefore likely that further disease-specific or opposite-effect variants remain to be identified. While the approach used here emphasizes shared genetic impact more than distinct genetic impact, we note that most shared-impact loci appeared in the original asthma or CRSwNP-specific GWASs. While most known types of asthma and CRS cases are represented in this study sample, early childhood asthma is underrepresented. The analysis of CRSsNP in the UKB dataset was underpowered due to underrepresentation, and the genetics of CRSsNP requires further study. The observational approach provides strong correlations and the sophisticated measurement tools provide strong evidence, from which establishing causal relationships require further study with potential benefits to treatment. The use of British and Finnish ancestries also limits analysis to variants represented in both ancestries, omitting several ancestry-enriched variants that require further study to determine their validity. This also limits the impact of these results outside of European ancestries[59]. The genomic lambda values suggest inflation is likely attributable to a polygenic signal, although it may also be partially influenced by residual population stratification not fully controlled by principal component adjustments.

On the whole, this study shows that shared heritable affinity to developing CRSwNP has an even higher overlap with that of asthma than expected. Shared heritability to asthma and CRSwNP is explained in part by type 2 inflammation, as previously noted, but also points to other immunological dimensions, evidenced by overlaps with allergic and atopic phenotypes, immune deficiency, and autoimmunity. CRSwNP-specific heritability, carried in part by rare disease-specific variants such as the *TP63* missense variant rs190865056-A, seems to differ from shared heritability, at least in its lack of impact on allergic phenotypes, while asthma-specific heritability shows few differing comorbidities. Given the success of biologic therapeutics in CRSwNP, asthma, and atopic dermatitis, it is likely that further insight can be obtained by future investigations combining other immunologically related endpoints with asthma or CRSwNP.

## Methods
### Study sample
Finnish participants were obtained from the FinnGen study[60], a recently completed nation-wide collection of Finnish genetic samples comprising close to 10% of the population. The dataset used in this study comprised 392,423 participants from FinnGen release 9 (February 2022), including both legacy samples, maintained by the Biobank of the Finnish Institute for Health and Welfare (THL), and recent biobank samples recruited at Finnish university hospitals. The sample population evenly represents both sexes (55.9% females) and all ages. Phenotypic information include disease information from the Care Register for Health Care (THL), the Primary Health Care Register (THL), the Causes of Death Register (Statistics Finland) and the Drug Reimbursement Register (KELA, the Social Insurance Institution of Finland). Asthma cases were established by a diagnosis of J45 (ICD-10) or 493 (ICD-9 or ICD-8) in the Care Register for Health Care or Causes of Death Register. CRSwNP cases were established by a diagnosis of J33 (ICD-10), 471 (ICD-9), or 505 (ICD-8) in these registers. CRSsNP cases were established by a diagnosis of J32 (ICD-10), 473 (ICD-9), or 503 (ICD-8) in these registers in the absence of nasal polyposis diagnosis (J33 in ICD-10, 471 in ICD-9 and 505 in ICD-8).

The co-occurrence of phenotypes is summarized in Supplementary Tables 6 and 7. For comparative effect analyses, we used atopic dermatitis (n = 13,473), allergic phenotypes (allergic rhinitis, allergic conjunctivitis, or allergic asthma; n = 34,614), and autoimmune diseases

($n$ = 96,150). These endpoints are encoded in FinnGen R9 PheWEB as L12_ATOPIC, ALLERG_RHINITIS, H7_ALLERGICCONJUCTIVITIS, ALLERG_ASTHMA, and AUTOIMMUNE, respectively.

For further sub-phenotype analysis, we selected participants with asthma diagnosed before age 16 ($n$ = 5865, 13.9 %), as defined in the FinnGen R9 PheWEB (ASTHMA_CHILD_EXMORE). Asthma-related infections were defined as specialist-diagnosed upper and respiratory tract infections in asthma cases ($n$ = 16,018, 37.9 %) as defined in FinnGen R9 PheWEB (ASTHMA_INFECTIONS).

Participants in FinnGen provided informed consent for biobank research, based on the Finnish Biobank Act. Alternatively, separate research cohorts, collected prior to the Finnish Biobank Act came into effect (in September 2013) and the start of FinnGen (August 2017), were collected based on study-specific consents and later transferred to the Finnish biobanks after approval by Fimea, the National Supervisory Authority for Welfare and Health. Recruitment protocols followed the biobank protocols approved by Fimea. The Coordinating Ethics Committee of the Hospital District of Helsinki and Uusimaa (HUS) approved the FinnGen study (protocol HUS/990/2017). The FinnGen study approval permits are listed in Supplementary Table 4, and biobank access decisions are listed in Supplementary Table 5.

UK participants were obtained from the UKB[61]; a prospective cohort with extensive phenotype and genotype data from up to 500,000 individuals aged 40–69 at recruitment. The phenotype definitions were based on ICD10 codes recorded from the hospital inpatient records from the British health registries comprising both primary and secondary diagnoses. These data have been coupled with the UKB and can be found in fields 11634–11846. CRSwNP is defined by ICD-10 code J33 (Nasal polyp). CRSsNP is defined by ICD-10 code J32 (Chronic sinusitis), excluding individuals with the diagnosis code J33. Asthma is defined by ICD-10 codes J45 (Asthma) and J46 (Status asthmaticus).

For controls, we selected all study participants without any of the three conditions.

## Genotyping and quality control
As described above, a total of 392,423 FinnGen samples were collected both as legacy cohorts that have been genotyped previously, and recruited samples genotyped after recruitment, using a total of 15 different DNA chips. We excluded variants with a high missingness (>2%), minor allele count <5, and considerable violation of Hardy-Weinberg equilibrium (pHWE < 1e-6). A detailed description of genotyping chips used, quality control, and imputation have been previously described elsewhere[60]. We excluded samples with ambiguous sex, genotype missingness >0.05, incomplete minimum phenotype information (age, sex, and hospital record data), and non-Finnish ancestry in principal component analysis. For FinnGen release 9, the SISu[62] v4 LD reference panel was used in imputation, resulting in 20,167,810 variants analyzed over 377,277 participants.

For UKB analysis, the genotyping and QC pipeline has been described previously. On top of this QC pipeline, for UKB we have chosen to include only variants with MAF > 1% for computational and statistical reasons. This resulting in the inclusion of 7,268,980 variants. Functional variants detected in FinnGen were separately tested for replication in the UKB with no constraint on allele frequency.

## Genome-wide association
We analyzed mixed logistic model regression genome-wide using the REGENIE[63] software (v2.2.4) on 20,167,811 variants in FinnGen. Covariates used in all FinnGen GWAS were sex, age, genotyping batch, and principal components (PCs) 1–10.

In UKB, the analysis was performed in European unrelated individuals with logistic regression using plink2.0[64]. Covariates in all UKB analyses were sex and PCs 1–15.

Principal components were computed with fastPCA, which is also described in the QC pipeline. For a detailed description of principal component analysis in the FinnGen cohort, please see the pipeline open source pages (https://github.com/FINNGEN/). Only samples with Finnish ancestry in PC analysis merged with 1000 Genomes[65] samples were included. The post-QC number of participants included in the analyses was 377,277.

GWAS results from FinnGen and UKB were meta-analyzed for validity. Only meta-analyzed variants with $p$ < 5e-8 were considered significant.

Genetic loci were selected starting from the variant with the lowest $p$-value (lead) in the chromosome, including variants with LD > 0.1 of the lead variant within a 2 megabase window around the lead variant, then beginning anew until no variants with $p$ < 5e-8 remained. We defined loci as novel if they did not appear in Open-Targets or GWAS Catalog with an association to the associated phenotype. We also manually curated loci according to the phenotypes of asthma[10,17,18,23,24], CRSwNP[9,12] or CRSsNP[12].

Loci were fine-mapped for credible sets of causative variants using the SuSiE[30] software v0.9.26 in FinnGen. All credible set variants were annotated for predicted functional consequence. Fine-mapped variants with non-synonumous annotations were considered replicated only if they were GWS in meta-analysis with UKB.

Colocalization of finemapped credible sets were investigated using an in-house pipeline powered by eCAVIAR[66]. High-quality credible sets from the FinnGen R9 results were colocalized with allergic phenotypes (see above), in addition to eQTLs of gene expression in non-gonadal tissue, as made available by GTEx[34]. A colocalized signal was considered indicative of a truly shared loci if posterior probability was higher than 90 %.

## Cross-trait analysis
In order to increase our yield among shared genetic contributors to both asthma and CRS, we ran a GWAS using all cases with asthma, CRSwNP or CRSsNP as case. Quality control and analysis pipelines were the same as for single-phenotype analyses. Asthma and/or CRS cross-trait analysis was run separately in FinnGen and UKB, and cohort-specific analyses were then meta-analyzed as described below.

## Meta-analysis
We conducted four sets of inverse variance-weighted meta-analyses, including one for each single-phenotype GWAS and one for the cross-trait GWAS. The meta-analyses were performed using METAL V.2011-03-25[67]. We assessed statistical significance using the standard genome-wide significance level ($p$ < 5e-08).

## Bayesian analyses
In order to specify whether loci identified in the asthma and/or CRS analysis were truly shared, or whether an association was carried by a single phenotype only, we employed a Bayesian analysis framework (MetABF[26]). This analysis pipeline employs a covariance matrix to adjust for overlapping case counts, and approximates a Bayesian factor (ABF) for different explanatory models of a given set of effect sizes for a given variant. We ran the Bayesian analysis on asthma and/or CRS cross-trait GWAS lead variants. Analysis was run in FinnGen and UKB separately, since case overlaps could only be adjusted for within cohorts. The tested models in analyses were:

NULL, where there is no true effect on the tested phenotypes

SHARED, where all phenotypes had one identical or highly similar true effect size;

ASTHMA, where the true effect for asthma is non-zero and 0 for all other phenotypes; and

CRSwNP, where the true effect for CRSwNP is non-zero and 0 for all other phenotypes.

NULL and phenotype-specific models are calculated using single models. The SHARED model is calculated by evaluating the ABF for one model where the effect is exactly the same in both CRSwNP and Asthma ("fixed"), and another where the effect is highly correlated (at least 90%; "correlated"). Priors for "fixed" and "correlated" are set to half that of NULL and phenotype-specific models to adjust for an otherwise inflated ABF. These two Bayesian factors are then summed to establish an ABF for the SHARED model, which then represents a hypothesized shared effect. A model was considered confirmed if it reached a posterior probability of 80%. No independent signal model was considered, as the choice of using shared controls would deflate signals from variants with opposing effects.

To analyze trends of shared and distinct impact on asthma and CRSwNP, we employed a recently developed approach (linemodels[29]) to group variants according to estimated effect. Briefly, this approach performs a probabilistic clustering of variants based on effect sizes, correcting for epidemiological overlap between cases of different phenotypes specified for subpopulations (FinnGen and UKB). We tested for grouping according to three models: "ASTHMA", where variants were assumed to have a non-zero effect on asthma and not CRSwNP; "CRSwNP", where variants were assumed to have a non-zero effect on CRSwNP and not asthma; and "BOTH", where variants were assumed to have a highly correlated and proportional effect to both asthma and CRSwNP. The threshold for belonging to a particular cluster was set at posterior probability >90%.

### Multi-marker Analysis of GenoMic Annotation

We used the Multi-marker Analysis of GenoMic Annotation (MAGMA[35], v1.10) software to run gene, gene set, and tissue enrichment analyses from meta-analyzed summary statistics data from CRSwNP, Asthma, and Asthma and/or CRS analyses. Genes were downloaded from NCBI RefSeq (n = 37358). Significance threshold was set at $p < 4.46e\text{-}7$ for the gene analysis. For gene set analysis, we used canonical gene sets (n = 6490) described at MSigDB (v2023.1.Hs), with a significance threshold of $p < 7.70e\text{-}6$. Tissue enrichment analysis was performed using the publicly available FUMA pipeline[68] and leveraged RNA-seq data summaries provided by GTEx v8[34]. This analysis tests for associations between tissue specific gene expression profiles, and disease-gene associations. 54 GTEx tissues were used and three phenotypes tested, setting the significance threshold at $p < 3.09e\text{-}4$.

### Reporting summary

Further information on research design is available in the Nature Portfolio Reporting Summary linked to this article.

## Data availability

The summary statistics from genome-wide meta-analysis generated in this study have been deposited in the GWAS Catalog database under accession codes GCST90652530 (asthma), GCST90652531 (CRSwNP), GCST90652532 (CRSsNP) and GCST90652529 (asthma and/or CRS) [https://www.ebi.ac.uk/gwas/]. The summary statistics from specific cohorts are publicly available at locally hosted buckets (FinnGen: https://storage.googleapis.com/fg-publication-green-public/F_2021_032_20250820/ASTCRSNP.tar.gz and UKB: https://storage.googleapis.com/fg-publication-green-public/F_2021_032_20250820/ASTCRSNP_ukb.tar.gz). Based on National and European regulations (GDPR) access to individual-level sensitive health data must be approved by national authorities for specific research projects and for specifically listed and approved researchers. The health data described here was generated and provided by the National Health Register Authorities (Finnish Institute of Health and Welfare, Statistics Finland, KELA, Digital and Population Data Services Agency) and approved, either by the individual authorities or by the Finnish Data Authority, Findata, for use in the FinnGen project. Therefore, we, the authors of this paper, are not in a position to grant access to individual-level data to others. However, any researcher can apply for the health register data from the Finnish Data Authority Findata (https://findata.fi/en/permits/) and for individual-level genotype data from Finnish biobanks via the Fingenious portal (https://site.fingenious.fi/en/) hosted by the Finnish Biobank Cooperative FINBB. All Finnish biobanks can provide access for research projects within the scope regulated by the Finnish Biobank Act, which is research utilizing the biobank samples or data for the purposes of promoting health, understanding the mechanisms of disease or developing products and treatment practices used in health and medical care. You can learn more about accessing other FinnGen data here: https://www.finngen.fi/en/access_results. Access to UKBB data can be obtained upon request [http://www.ukbiobank.ac.uk/resources/]. Variant summary data for the SISu v4.1 reference panel are available at https://sisuproject.fi with information on previous versions available upon request. Gene annotations and gene sets were downloaded from public repositories (NCBI [https://www.ncbi.nlm.nih.gov/refseq/], MSigDB [https://www.gsea-msigdb.org/gsea/msigdb/collections.jsp]). RNA expression data summaries for tissue enrichment analysis are publically available at the GTEx consortium [https://www.gtexportal.org/home/].

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

## Acknowledgements

We want to acknowledge the participants and investigators of the FinnGen study. The FinnGen project is funded by two grants from Business Finland (HUS 4685/31/2016 and UH 4386/31/2016) and the following industry partners: AbbVie Inc., AstraZeneca UK Ltd, Biogen MA Inc., Bristol Myers Squibb Inc. (and Celgene Corporation & Celgene International II Sàrl), Genentech Inc., Merck Sharp & Dohme LCC, Pfizer Inc., GlaxoSmithKline Intellectual Property Development Ltd., Sanofi US Services Inc., Maze Therapeutics Inc., Johnson&Johnson Innovative Medicine Inc., Novartis AG, Boehringer Ingelheim International GmbH and Bayer AG. Following biobanks are acknowledged for delivering biobank samples to FinnGen: Auria Biobank (www.auria.fi/biopankki), THL Biobank (www.thl.fi/biobank), Helsinki Biobank (www.helsinginbiopankki.fi), Biobank Borealis of Northern Finland (https://www.ppshp.fi/Tutkimus-ja-opetus/Biopankki/Pages/Biobank-Borealis-briefly-in-English.aspx), Finnish Clinical Biobank Tampere (www.tays.fi/en-US/Research_and_development/Finnish_Clinical_Biobank_Tampere), Biobank of Eastern Finland (www.ita-suomenbiopankki.fi/en), Central Finland Biobank (www.ksshp.fi/fi-FI/Potilaalle/Biopankki), Finnish Red Cross Blood Service Biobank (www.veripalvelu.fi/verenluovutus/biopankkitoiminta), Terveystalo Biobank (www.terveystalo.com/fi/Yritystietoa/Terveystalo-Biopankki/Biopankki/) and Arctic Biobank (https://www.oulu.fi/en/university/faculties-and-units/faculty-medicine/northern-finland-birth-cohorts-and-arctic-biobank). All Finnish Biobanks are members of BBMRI.fi infrastructure (https://www.bbmri-eric.eu/national-nodes/finland/). Finnish Biobank Cooperative—FINBB (https://finbb.fi/) is the coordinator of BBMRI-ERIC operations in Finland. The Finnish biobank data can be accessed through the Fingenious® services (https://site.fingenious.fi/en/) managed by FINBB. This work was supported by Academy of Finland (grant no. 360387 to E.C.S., grant no. 338229 to E.S.), and the Finnish Medical Foundation (grant no. 6465 to E.C.S.). The Academy of Finland Center of Excellence for Complex Disease Genetics (grant no 312074 and 352793 to A.P.), Sigrid Juselius Foundation, Finland (to A.P.), Instand-NGS4P, EU H2020, grant no 874719, REALMENT, EU H2020, grant no 964874 (to A.P.) We thank the GWAS catalog (https://www.ebi.ac.uk/gwas/) and the GTEx database (https://www.gtexportal.org/) for sharing expression data and summary statistics of GWAS studies. This research has been conducted using the UK Biobank Resource under Application Number 54273. Open access funded by Helsinki University Library.

## Author contributions

Each author's contribution(s) to the paper is listed according to the CRediT model. Conceptualization: E.C.S. Methodology: E.C.S., K.F.R., Ee.S. Validation: E.C.S. Formal analysis: E.C.S., K.F.R., Ee.S. Investigation: E.C.S., K.F.R., Ee.S., FINNGEN. Resources: K.B., FINNGEN. Data Curation: E.C.S., K.F.R., FINNGEN. Visualization: E.C.S. Funding acquisition: E.C.S., A.P., A.M. Project administration: A.P., A.M. Supervision: A.M., A.P., K.B., J.K. Writing—original draft: E.C.S. Writing—review and editing: E.C.S., K.F.R., Ee.S., A.B.Z., T.L., S.T.S., H.K., J.K., K.B., A.P., A.M.

## Competing interests

The authors declare no competing interests.

## Additional information

[1]Department of Otorhinolaryngology—Head and Neck Surgery, University of Helsinki and Helsinki University Hospital, Helsinki, Finland. [2]Institute for Molecular Medicine Finland FIMM, HiLIFE, University of Helsinki, Helsinki, Finland. [3]Research Program in Systems Oncology, Faculty of Medicine, University of Helsinki, Helsinki, Finland. [4]COPSAC, Copenhagen Prospective Studies on Asthma in Childhood, Copenhagen University Hospital – Herlev and Gentofte, Copenhagen, Denmark. [5]Center for Life Course Health Research, Faculty of Medicine, Oulu, Finland. [6]Biocenter Oulu, University of Oulu, Oulu, Finland. [7]Medical Research Center Oulu, Oulu University Hospital and University of Oulu, Oulu, Finland. [8]Department of Otolaryngology, University of Tampere, Tampere, Finland. [9]Wellbeing Services County of Pirkanmaa, Pirkanmaa, Finland. [10]Department of Otorhinolaryngology, University of Eastern Finland, Joensuu and Kuopio, Kuopio, Finland. [11]Wellbeing Services County of Pohjois-Savo, Kuopio, Finland. [12]Department of Allergology, Inflammation Center,

Helsinki University Hospital and University of Helsinki, Helsinki, Finland. [13]Krefting Research Centre, Institute of Medicine, Department of Internal Medicine and Clinical Nutrition, University of Gothenburg, Gothenburg, Sweden. [14]Department of Respiratory Medicine, Seinäjoki Central Hospital, Seinäjoki, Finland. [15]Tampere University Respiratory Research Group, Faculty of Medicine and Health Technology, Tampere University, Tampere, Finland. [16]Department of Clinical Medicine, University of Copenhagen, Copenhagen, Denmark. [17]Stanley Center for Psychiatric Research, Broad Institute of MIT and Harvard, Cambridge, MA, USA. [18]Analytic and Translational Genetics Unit, Department of Medicine, Department of Neurology and Department of Psychiatry, Massachusetts General Hospital, Boston, MA, USA. ✉e-mail: elmo.saarentaus@helsinki.fi

## FinnGen

**Elmo C. Saarentaus** [1,2,3] ✉, **Eeva Sliz**[4,5,6], **Argyro Bizaki-Vallaskangas**[7,8], **Tarja Laitinen**[2], **Sanna Toppila-Salmi**[9,10,11], **Hannu Kankaanranta**[12,13,14], **Johannes Kettunen**[4,5,6], **Aarno Palotie**[2,15,16] & **Antti Mäkitie** [1,3]

A full list of members and their affiliations appears in the Supplementary Information.

