## [Transparent Peer Review file · Nature Communications]

131 genetic loci highlight immunological pathways and tissues in nasal polyposis and asthma

Corresponding Author: Dr Elmo Saarentaus

Version 0:

Reviewer comments:

Reviewer #1

(Remarks to the Author)

This study sought to evaluate the genetics of CRSwNP, CRSsNP and asthma in two large cohorts by performing genome-wide meta-analyses of these traits. Cross-trait comparisons and testing for phenotype sharing of lead SNPs at loci were also performed. As written, the manuscript is dense and the context for many of the analyses is missing. The overall impact of this manuscript is limited by these being a series of genetic analyses without any mechanistic results.

Beginning on lines 89-90, and repeated throughout the manuscript, is the description of genes representing loci. The genetic analyses being conducted are association tests between genetic variants (not genes) and traits. Referencing a locus by a gene implies that this gene is causally associated with a trait, which is not the case. There are other genes at each of these loci. Fine to say that loci have been associated and that type 2 inflammatory genes are among the genes in these regions, but there has to be an acknowledgment that other genes are present as well. This needs to be corrected throughout the manuscript and figures where gene names should be replaced with chromosomal band numbering or the rsID of the lead SNP at a locus.

In lines 95-97, the authors cite their previous study (reference #17) for demonstrating genetic correlation. However, this reference does not provide any data on asthma or CRSwNP.

Why was age corrected in the GWASs? Asthma heritability is greater for childhood-onset asthma than it is for adult-onset asthma and more loci have been associated in the former. What effect does the age correction have on the results?

The 12q13.11 locus that includes HDAC7 is not novel and has been reported in multiple prior asthma GWASs (Pividori et al, Lancet Resp, 2019, Ferreira et al AJHG, 2019, Tsuo et al. Cell Genomics, 2022). Why is this being reported as novel here? There needs to be more clarity and details about how novelty was determined for these SNPs. I am concerned that other lead SNPs could also be misclassified as novel in this manuscript.

Additional information about the subjects should be included in Table 1. At a minimum, age and sex for each group should be reported in Table 1.

Where are the SuSie fine-mapping results (number of credible sets at each locus, number of SNPs per credible sets)? These results are likely to be some of the more interesting analyses in this study, but these results are not discussed. Instead, it is assumed (and not mentioned) that the causal SNPs at loci are protein-modifying variants. However, this assumption is not justified. SNPs could also be eQTLs and modify the expression level of genes and proteins.

Neither Figure 1 nor Figure 2 are mentioned in the results section.

The interpretations of Figure 3 and Figure 4 are very confusing and not described well in the manuscript. From what I see, the Bayesian model is indicating that most SNPs are shared, and this is consistent with the point estimates and overlapping confidence intervals on the right part of the figures. However, these results are described in the figures and manuscript to mean that these signals associate with asthma.

What are the dashed lines in Figure 5 indicating?

A major limitation that is not discussed is that these studies were restricted to subjects identified as European and do not include ancestrally diverse populations.

The first paragraph of the results (lines 110-115) should be moved to the methods.

Reviewer #2

(Remarks to the Author)

Reviewer #3

(Remarks to the Author)

This large study tackles a well-constructed hypothesis using appropriate methodology and has some novel findings, e.g. that CRSwNP specific loci does not associate with allergic phenotypes, and novel non-synonymous associations.

My biggest comment is the lack of results presented between genetic overlap with CRsNP (with both asthma and CRSwNP) in the Bayesian grouping, despite the novel loci being detected including CRSsNP in the case definition. (To not distract from the main results, it may be most appropriate to add these results to the supplementary material.) There are fewer genome-wide significant loci with the CRSsNP specific GWAS, despite larger sample size (probably due to less heritability), and contrasting genetic overlap with CRsNP would likely be a useful null comparison, providing stronger support for the overlap between asthma and CRSwNP.

My remaining comments are mostly editorial/requests for clarification.

Specific comments – Abstract

- Lines 44-45: “These loci largely cluster into three groups, with a shared risk being the most frequent.” – I am not sure what this means, or what result this refers to (perhaps the metABF groups?).

Specific comments – Introduction

- Line 62-63: “CRSwNP is underdiagnosed in the general population, and the true prevalence is likely higher” – is it possible to give a citation for this statement?

- Line 67: “The majority of Western CRSsNP and CRSwNP cases” – perhaps rephrase as “The majority of CRSsNP and CRSwNP cases in Western countries”

- Lines 95-96: “Our previous in silico analyses using LD score regression [17] established a genetic correlation of 68.7% between asthma and CRSwNP” – ref 17 is the LDscore regression paper; what is the reference for the previous in silico analyses?

Specific comments – Results

- General comment (which pertains to missing information in the methods section but I was curious about this as I read the results) – how did the authors decide when an association was novel vs known? Did they e.g. use GWAS catalog, or literature? If the former, which search terms did they use? Did they match SNPs exactly, or did they allow for a window around SNPs?

- Lines 114-115: “GWAS results from FinnGen and UKB were meta-analyzed for validity and replication” – This wording does not make sense to me (this is not replication in the traditional sense where there is a very clear discovery and replication cohort) – do you rather mean to assess heterogeneity of effects between cohorts?

- Lines 120-121: “While four asthma lead variants had a heterogeneous effect estimate ($I^2 > 50\%$)” – in table 2, only 2 variants had $I^2 > 50\%$ (rs114425738-G and rs11168249-C), not 4 variants as stated.

- Lines 121-122: Weird line break

- Lines 132-137: From the wording, it is not clear which one of the associations are novel

- Line 149: “However, as the intercept values” should be “However, as the LD score regression intercept values”

- Line 152: “the expected associations” should be “the expected distribution”

- Lines 155-160: It is probably also worth pointing out that the other advantage of using MetABF is the ability to compare effects across phenotypes with varied sample size

- Lines 183-193: While I agree that this is an interesting observation that should be pointed out, I think it should be attenuated somewhat: While the CRSwNP point estimates are mostly higher in Figure 5, the confidence intervals of most overlap with asthma. In addition, the closing sentence noting the disease-specific overlap in findings are based on loci that crossed genome-wide significance, which is limited by the fact that the disease-specific GWAS are of different sample size.

- Characterization of loci section: Suggest splitting it out into two separate sections, one focused on related phenotype associations, and one focused on fine-mapping

- Line 226: Suggest dropping “considered” (either the association crossed the threshold, or not)

- Line 256-258: Could this observation perhaps be due to underrepresentation of asthma relevant cells in the GTEx lung tissue?

- Line 266: “overlapped 100%” – it was not clear to me what this meant – perhaps restate this as the confidence interval upper bound reaching 100%. In addition, for supplementary table 3 etc, correlation coefficients should be expressed as a

proportion, not a percentage

- Line 270-271: A couple of brief sentences stating the previous method that was used, and why it led to inflated estimates, would be appreciated.

Specific comments – Discussion

- Line 273: The grammar of the opening sentence can be improved.

- Line 278: Suggest describing ancestry as European ancestry rather than Caucasian (PMID: 35212875)

- Line 299: Suggest “differ from” instead of “distinguish from”

- Line 353: “suggest inflation likely” should be “suggest inflation is likely”

Specific comments – Materials and Methods

- Lines 375-380: The phenotype definitions are more broad than the criteria stated on lines 110-112; suggest updating lines 110-112 accordingly

- Line 381: Supplementary Data 7 seems to be an author list

- Line 440: “CRSwNP or CRSwNP” should be “CRSwNP or CRSsNP”

- Lines 503-506: MetABF software details are missing

Version 1:

Reviewer comments:

Reviewer #1

(Remarks to the Author)

The authors have conducted additional analyses and revised the manuscript to my satisfaction. One recommendation would be to edit Figure 1 so that it is more illustrative; consider making the points larger and including breaks in the y-axis to reduce the white space.

Reviewer #2

(Remarks to the Author)

I am pleased with the authors' response and commend their thorough investigation of GWAS and co-morbidity, which presents a significant challenge.

Reviewer #3

(Remarks to the Author)

No additional comments; the revision adequately addressed my previous comments.

Response to reviewer comments

Reviewer #1 comments:

This study sought to evaluate the genetics of CRSwNP, CRSsNP and asthma in two large cohorts by performing genome-wide meta-analyses of these traits. Cross-trait comparisons and testing for phenotype sharing of lead SNPs at loci were also performed. As written, the manuscript is dense and the context for many of the analyses is missing. The overall impact of this manuscript is limited by these being a series of genetic analyses without any mechanistic results.

- We thank the Reviewer for raising the issues of density and explanation. In order to address this, we have thoroughly revised the language of the manuscript.

Beginning on lines 89-90, and repeated throughout the manuscript, is the description of genes representing loci. The genetic analyses being conducted are association tests between genetic variants (not genes) and traits. Referencing a locus by a gene implies that this gene is causally associated with a trait, which is not the case. There are other genes at each of these loci. Fine to say that loci have been associated and that type 2 inflammatory genes are among the genes in these regions, but there has to be an acknowledgment that other genes are present as well. This needs to be corrected throughout the manuscript and figures where gene names should be replaced with chromosomal band numbering or the rsID of the lead SNP at a locus.

- This is a good point. We recognize that the way this was represented in our original submission may mislead the reader, and have revised to refer to loci based on their cytobands. In addition, we have revised throughout when discussing loci to separate genes. We have also added the lead variants into Figures 3 & 4, along with locus.
 - o Lines 90–91: “In asthma, over 212 genomic loci have been identified to date, including **loci harboring GSDMB/ORMDL, HLA, TSLP, IL1RL1/IL18R, and IL33.**”
 - o Lines 118–119: “...although heterogeneity for **1p22.1 (harboring TGFBR3)** was high ($I^2 = 76.3\%$).”
 - o Lines 119–121: “Of the asthma-associated loci, 26/87 were GWS associated with CRSwNP, including three (**5q22.1, 6p21, and 9p24.1, near genes TSLP, HLA, and IL33, respectively**) associated with CRSsNP as well.”
 - o We also provide evidence of a trend of shared loci having a higher absolute risk increase for CRSwNP, including many loci traditionally considered asthma-specific, such as **5q22.1 near TSLP, or 9p24.1 near IL33.**

In lines 95-97, the authors cite their previous study (reference #17) for demonstrating genetic correlation. However, this reference does not provide any data on asthma or CRSwNP.

- Thanks for noticing this, we apologize for the missed reference (which should be #10; LD Score regression is referenced in #17). We have amended accordingly.
 - o Lines 96-98: “Our previous in silico analyses (**12**) using LD score regression (**19**) established a significant genetic correlation of 68.7% between asthma and CRSwNP [...]”.

Why was age corrected in the GWASs? Asthma heritability is greater for childhood-onset asthma than it is for adult-onset asthma and more loci have been associated in the former. What effect does the age correction have on the results?

- We have now rerun the FinnGen asthma analysis without the age covariate. In our analysis, the results replicate without noticeable difference. We have added Supplementary Figure 4, along with

the following note: “Interestingly, age correction did not appear to have any impact in FinnGen analysis (Supplementary Figure 4)”.

To answer the question specifically: FinnGen and the UK Biobank have recruited participants across a broad range of ages, thus providing an avenue of confounding due to differences in possible time to be diagnosed with asthma. With this in mind, it was feared that not correcting for age may inflate genetic associations to genetic drift between generations. The comparison is now included in the manuscript as a sensitivity analysis (supplementary figure 1).

- Results, lines 122 – 124: “Interestingly, age correction did not appear to have any impact in the FinnGen analysis (Supplementary Figure 1).”

The 12q13.11 locus that includes HDAC7 is not novel and has been reported in multiple prior asthma GWASs (Pividori et al, Lancet Resp, 2019, Ferreira et al AJHG, 2019, Tsuo et al. Cell Genomics, 2022). Why is this being reported as novel here? There needs to be more clarity and details about how novelty was determined for these SNPs. I am concerned that other lead SNPs could also be misclassified as novel in this manuscript.

- We apologize for this oversight, and thank the Reviewer for catching this. We have revised the number of previously unidentified asthma loci to ten. Given the vast literature on genome-wide associations of asthma, we agree the need for further specificity on methods here. For clarification, we have now added a note on this in the Methods section.
 - Lines 473–476: “We defined loci as novel if they did not appear in OpenTargets or GWAS Catalog with an association to the associated phenotype. We also manually curated loci according to the phenotypes of asthma [17,18,23,24,67], CRSwNP [9,12] or CRSsNP [12]”.
 - In addition, the number of new asthma loci has been amended accordingly. The lines on HDAC7 have been removed.

Additional information about the subjects should be included in Table 1. At a minimum, age and sex for each group should be reported in Table 1.

- We thank the reviewer for raising this concern, and have amended Table 1 accordingly.

Where are the SuSie fine-mapping results (number of credible sets at each locus, number of SNPs per credible sets)? These results are likely to be some of the more interesting analyses in this study, but these results are not discussed. Instead, it is assumed (and not mentioned) that the causal SNPs at loci are protein-modifying variants. However, this assumption is not justified. SNPs could also be eQTLs and modify the expression level of genes and proteins.

- We thank the reviewer for raising this concern, which parallels with how the loci are presented. The credible sets of causal variants have now been detailed in Supplementary Data 6–9, and a colocalization pipeline has been added that incorporates shared signal and eQTL analysis (Supplementary Data 11).
 - Lines 253–259: “Thirdly, we colocalized ($PP > 0.9$) credible sets with eQTLs identified in gene expression data in 44 non-gonadal tissues from GTEx [34] (Supplementary Data 11). 16 loci colocalized with differing expression of 29 genes among all 44 tissues. In lung tissue samples, three loci colocalized with gene expression of three genes: 4q24 (near *NPNT*) had codirectional effect with expression of long non-coding RNA (lncRNA) ENSG00000250740; 12q13.2 (near *RPS26*) had opposite effect on *SUOX* and *RPS26* expression; and 17q21.1 (near *GSDMB*) had codirectional effect with *MED24* expression.”

Neither Figure 1 nor Figure 2 are mentioned in the results section.

- We thank the reviewer raising this concern, and have amended accordingly.
 - Figure 1 is now referenced in line 115, and Figure 2 is referenced in lines 116 and 125.

The interpretations of Figure 3 and Figure 4 are very confusing and not described well in the manuscript. From what I see, the Bayesian model is indicating that most SNPs are shared, and this is consistent with the point estimates and overlapping confidence intervals on the right part of the figures. However, these results are described in the figures and manuscript to mean that these signals associate with asthma.

- We thank the reviewer for raising this concern, which may correlate with the stated density of the text. As this analysis seeks to describe what loci found in the asthma and/or CRSwNP GWAS are truly shared and which merely carried by the most common phenotype (asthma), this description is now seen as incomplete. We have made the following edits:
 - o Results, Bayesian phenotype grouping, lines 165–173: **“Through this analysis, we identified that 13 of the 17 loci, which did not appear in single-phenotype GWAS, contributed to both asthma and CRSwNP. [...] Of the remaining 4 loci, three appeared to affect only asthma: 2q24.1 near *GPD2*, 8p23.1a near *SGK223*, and 8p23.1d near *DEFB136* (Figure 3). The impact of the 3p14.3 locus near *PPA1P16* could not be definitively categorized as either shared or asthma-specific. Among the shared impact loci was 7q22.3, lead by the missense variant rs6967330-G in the viral receptor-encoding *CDHR3*; this variant has been previously linked to early childhood asthma with severe exacerbations[27] and CRS[28].”**

What are the dashed lines in Figure 5 indicating?

- We thank the reviewer for catching this, the caption of Figure 5 now includes: **“All three models have an assumed correlation of 0.999, with dashed lines representing expected deviation with the correlation.”**

A major limitation that is not discussed is that these studies were restricted to subjects identified as European and do not include ancestrally diverse populations.

- We thank the reviewer for raising this important limitation. The limitations section now added: **“The use of British and Finnish ancestries also limits analysis to variants represented in both ancestries, omitting several ancestry-enriched variants that require further study to determine their validity. This also limits the impact of these results outside of European ancestries [60].”** with an added reference to an excellent paper by AR Martin et al, “Human Demographic History Impacts Genetic Risk Prediction across Diverse Populations” published in AJHG in 2017.

The first paragraph of the results (lines 110-115) should be moved to the methods.

- We thank the reviewer for raising this concern, and have revised accordingly. The first sentence of the paragraph is left as a topic sentence for the following paragraph, with concurrent sentences removed (one sentence repeating information elsewhere in Methods) or moved to Methods.

Reviewer #2 comments:

In the manuscript titled: “131 genetic loci highlight immunological pathways and tissues in nasal polyposis and asthma” Elmo Saarentaus and coworkers show for the first time a genome-wide meta-analysis of asthma and CRSwNP in a very large number of clinical samples from the Finnish and British populations. They detect 17 new loci associating with asthma, 33 new loci associating with CRSwNP, and one locus associating with CRSsNP. Comparing effect estimates allows the authors distinguish a total of 131 loci, of which 20 associate only with asthma, 14 only with CRSwNP, and 71 loci with both.

Elmo Saarentaus and coworkers adopted an approach of combining cases from the three disorders (asthma, CRSsNP and CRSwNP) to emphasize shared biology, and shared genetic impact, more than distinct genetic impact, and suggests that most shared-impact loci also appear in the original asthma or CRSwNP-specific GWASs performed in this study.

In summary,

- What are the noteworthy results?

The authors distinguish a total of 131 loci, of which 20 associate only with asthma, 14 only with CRSwNP, and 71 loci with both. Noteworthy is the overlap, which might also be overestimated.

Will the work be of significance to the field and related fields? How does it compare to the established literature? If the work is not original, please provide relevant references.

Yes, this work will add significant knowledge to the field.

Does the work support the conclusions and claims, or is additional evidence needed?

Some conclusions are supported by the data while some might be slightly different if the analysis was done differently.

Are there any flaws in the data analysis, interpretation and conclusions? Do these prohibit publication or require revision?

The study focusses on shared-impact loci and has been performed very well, but I suggest the original asthma or CRSwNP-specific GWASs performed in this study are done differently.

Is the methodology sound? Does the work meet the expected standards in your field?

Yes

Is there enough detail provided in the methods for the work to be reproduced?

Some additional details in certain sections have been asked for in comments below.

Major comments:

One of the primary goals of this paper is to identify shared genetic factors and effects where the influence is either identical or very similar for both asthma and CRSwNP, as well as to pinpoint distinct effects where the genes or effects are unique to each condition. On row 356, the authors summarize the study's findings by stating: "On the whole, this study shows that shared heritable affinity to developing CRSwNP has an even higher overlap with that of asthma than expected". However, the authors do not separate the patients in the GWASs into distinct groups of those with asthma only and those with CRSwNP only. Instead, they include individuals with asthma in the CRSwNP GWAS analysis and those with CRSwNP in the asthma GWAS analysis. As shown in Figure 1: Study Overview and in Supplementary Figures 6 and 7, the GWAS of CRSwNP includes nearly 40% of individuals with asthma, and nearly 30% of those diagnosed with CRSwNP also have asthma (data from FinnGen only, as discussed below).

To more accurately identify genes specifically associated with CRS, I believe it would be preferable to conduct a GWAS on individuals diagnosed exclusively with CRS, as well as a GWAS on those with asthma only. The patients who have both CRSwNP and asthma will have genes linked to both conditions, and for this effect, the authors already have the combined GWAS data from 88,965 individuals. If the authors suggest that a genetic effect is seen for asthma but not for CRSwNP, or vice versa, shouldn't the asthma cases be excluded from the CRSwNP analysis, and vice versa? The findings and the identified disease-specific GWAS results—where 22.9% (20/87) of asthma loci were considered not to influence CRSwNP, and 28% (14/50) of CRSwNP loci were found not to significantly affect asthma—will likely change if the GWAS only include patients with one condition or the other, rather than both. This study also have more samples and the power lost will most likely be replaced by more specific results for each disease.

- We thank reviewer 2 for allowing us to elaborate on this issue. We appreciate the distinction raised here, which is central to the study design chosen.

In the case of CRS there is significant overlap with asthma (up to 50 %). To the problem it is hard to find a meaningful way of determining which cases are comorbid due to underlying shared biology. It is not known whether the comorbid condition is part of shared biology, or, admittedly less likely, a co-occurrence by chance. In other words, in the stated example, genetic effect to asthma but not CRSwNP does not necessarily mean protection from CRSwNP – even if the effect of a variant impacts asthma risk, carriers may still have co-morbid CRSwNP. Removing co-morbid participants will reduce power, and test for a different risk (asthma without CRSwNP) than what our main aim in this study is (asthma in general).

However, we have now run the analyses suggested by the reviewer. For instance, GWAS of FinnGen R9 CRSwNP cases without asthma detects 12 genomic loci, compared with the original 30 loci observed in FinnGen when no overlapping cases are removed – and all 12 loci are apparent in the original analysis. Similarly, asthma detects 30 loci compared with 33 loci in the original analysis, and CRSsNP highlights 4 loci vs 8 in the original FinnGen analysis. Using a cross-trait analysis that only considers participants with all three diagnoses (an intersection of all three) highlights 13 genomic loci, compared with 49 genomic loci when a union of all three is considered. This is likely due to a loss of power in the traditional approach. Despite these analyses excluding other diseases, they still show many of the same loci: 5q22.1b (near *TSLP*), 9p24.1 (near *IL33*), and 10p14b (near *GATA3*) all show as loci, with non-comparable effect estimates. We hold that drawing a line between overlapping and non-overlapping cases introduces a collider into the design, which our approach seeks to account for.

- Based on this feedback, it is evident that the study design has not been justified clearly enough in the original submission. In order to underscore this, we have made the following edits into the introduction section:
“Independent analyses of co-morbid disorders have been shown to have a potential to misrepresent shared etiologies [21]. Here, we leverage a previously established approach [11,16,22] using cross-trait analysis of all three endpoints to emphasize shared genetic risk, instead of limiting analysis to trait-specific associations. A note on the study design is included in the Supplement.”
- The suggested alternate analyses are also now referred to in the Supplementary Note:
“Multi-morbidity is a traditional problem in statistics, increasingly a problem of epidemiological studies in aging populations. Analysing diseases independently implies that the diseases can be considered independent. Common alternatives are multivariate modelling, or Bayesian hierarchies as used here.
- **“To investigate the challenge of co-morbidity, we also analysed the impact of selecting cases exclusively with one disorder in FinnGen R9, and ran their genome-wide association. We then compared these to our non-exclusive, “original” analyses.**
- **“Asthma without co-morbid CRS was observed for 36,116 of 42,163 participants (85.7 %), and a GWAS of these observed 30 loci, vs 33 in original FinnGen analysis. One of the 30 loci, at 3p24.1, is not observed in original FinnGen analysis. Variants at this locus were mostly absent in UKB, or showed no association ($p > 0.05$).**
- **“CRSwNP cases without comorbid asthma was observed for 3919 of 6255 cases (62.6 %), and a GWAS of these detected 12 genomic loci, vs 30 in original FinnGen analysis. 11 of the 12 loci is observed in the original FinnGen analysis, and one (6p25.3b) is not GWS in original analysis, but is observed in meta-analysis.**
- **“CRSsNP cases without asthma was observed for 9823 of 13,534 participants (72.6 %) and highlighting 4 loci vs 8 in the original FinnGen analysis. One of the 4 loci, at cytoband 2q14.1, is not observed in the original analysis, and contains a single rare variant (AF < 0.1 %) that is not tested for in the UKB.**
- **“Using a cross-trait analysis that only considers participants with all three diagnoses (an intersection of all three) highlights 13 genomic loci, compared with 49 genomic loci when a union of all three is considered. All 13 of these appeared in our analyses.**
- **“Notably, despite analyzing with an exclusive approach, we detect loci 5q22.1b (near *TSLP*) and 9p24.1 (near *IL33*) in all four analyses, and 10p14b (near *GATA3*) in both asthma and CRSwNP, albeit with non-comparable effect estimates. This demonstrates a loss of power in the categorical approach.”**
- To conclude, these additional analyses identified mostly the same, but a lower number of loci as our original analysis. Additionally, two loci not observed in the original analyses where genome wide

significant in the exclusive FinnGen analysis. These two loci were not replicated in the UKB analyses.

It is also noted in these tables that CRSsNP does not overlap with asthma diagnosis in the UK Biobank (UKB) samples, but there is a 30% overlap in FinnGen. This difference should perhaps be addressed earlier in the manuscript. In the discussion, the authors state, "CRSsNP also appeared underrepresented in the UKB, underpowering its analysis considerably." Would it be more accurate to say something like: "The analysis of CRSsNP in the UKB dataset was not possible because the ICD code for CRSsNP was not available in the population"?

- This is an important issue. After several inquiries with other researchers, and UK clinicians, we have yet to verify why CRSsNP is noted so infrequently in UK Biobank data. We have amended the suggested point in the limitations (lines 379–381) as follows: “The analysis of CRSsNP in the UKB dataset was underpowered due to underrepresentation, and the genetics of CRSsNP requires further study”.

Minor comments:

Rows 186-188 the authors write: “Notably, for loci associating with both asthma and CRSwNP, the effect on CRSwNP was consistently higher than that of asthma, with a slope of 1.88 in favor of CRSwNP in log-odds space.” Isn’t this just due to the fact that the CRSwNP sample size is smaller and that you only include genome-wide significant SNPs in the results?

- This is a complex issue to raise. For shared variants in particular, the genome-wide significance threshold does not readily reduce CRSwNP signals, as seen in e.g. the forest plots of figures 3 and 4. On the other hand, the smaller sample size in CRSwNP is due to diagnosis being made nearly solely by specialists, while asthma diagnoses in Finland are also made by primary care providers. And even if this was not the case, asthma is known to be more heterogeneous in clinical characteristics, and therefore might contain several subphenotypes with differing pathological mechanisms. We have added a point on this into the Discussion (lines 346–348):
 - o “The apparently higher CRSwNP impact among shared variants may well be explained by more special care-based diagnoses, in addition to a higher heterogeneity of asthma etiologies.”

In the section “Characterization of loci” the authors seem to be referring to specific associated SNPs or loci in general (and not whole genes). If by “loci” you mean “SNP” please specify this and if possible, include the rs-numbers and risk alleles together with each gene name mentioned.

- We thank the reviewer for raising this issue, which correlates with comments by reviewer 1. We have made edits to clarify when a locus is represented by its lead variant, and loci are no longer referred to by the genes nearest to lead variants.

In some places the authors write “nearest gene” and in some places just “gene” (also in supplementary data). Please specify if it is always nearest gene and if not specify why. Also in

supplementary data 6, some genes are written as Gene ID and other as Ensemble ID.

Corresponding NCBI RefSeq gene name would be good to include (since these gene names are used in every other table). It also would be informative to include top associated SNP in Supplementary Data 6.

- The first point coincides with that of reviewer #1's comment, which we appreciate and have amended accordingly. As for the second, we thank the reviewer very much for pointing out the inconsistency in coding, which revealed that our MAGMA analysis (previously in Supplementary Data 6) had unmatching gene definitions in UKB and FinnGen due to genome build difference that was not previously accounted for. The amended analysis identifies 620 gene-phenotype associations (previously 608). Re-analysis of an updated MAGMA pipeline also reveals 84 gene set-phenotype associations (up from previous 21), despite a stricter significance threshold. Our MAGMA analysis now reads (lines 261–275):
 - o “In order to further study the shared and distinct genes and pathways, we performed Multi-marker Analysis of GenoMic Annotation (MAGMA, v1.10)[35] to identify genes and gene sets enriched in each data set. This analysis summarizes GWAS results over gene regions to identify associated genes rather than single alleles. Genes were downloaded from NCBI RefSeq (n = 19,526). For gene set analysis, we used canonical gene sets (n = 7561) described at MSigDB 7.0[36]. MAGMA highlighted a total of 330 genes, of which 121 genes associated with CRSwNP, 7 associated with CRSsNP, and 246 associated with asthma (Supplementary Data X). MAGMA identified 246 genes in the asthma and/or CRS cross-trait analysis, of which 34 were not significant in single-phenotype MAGMA analyses. The gene set analysis identified 49 sets of genes with increased association to phenotypes (top 20 listed in Table 5), including known asthma-associated pathways such as Jak-STAT signaling[37, 38], innate lymphoid cell lines[39], and RUNX3 signaling[40]. CRSwNP was also associated with Jak-STAT and RUNX3 signaling, in addition to the NFAT transcription factor pathway and others. A notable shared signal was also confirmed with genes associated with atopic dermatitis mechanisms and therapies, a corollary to clinical manifestations[41–44].”

Row 177: “epidemiological overlap” is not completely clear what is meant. Could the authors please specify.

- We thank for raising this unconventional term. We have replaced “epidemiological overlap” with “correlation”.

Rows 206- The authors write “shared impact on CRSwNP and asthma were more often codirectional with other inflammatory endpoints. Atopic dermatitis had co-directional impact with 18.6 % (13/70) of shared loci.” Please specify if you only count genome-wide associated results or if you look at all SNPs in each case. It would be interesting to compare all of your 70 SNPs in which direction it is, if this was not done?

- We thank the reviewer for raising this. Our original analysis indeed only considered lead variants associating with a set of other phenotypes, but did not take into account whether GWAS with other phenotypes showed different loci. To take this into account, we used COLOC of credible sets of the associated loci.
- Lines 228–250: “In order to study the differences between loci groupings according to effect on asthma and CRSwNP, we also tested the impact of non-HLA loci (n = 130) credible sets on FinnGen R9 endpoints selected based on previously observed[15,16,20] asthma comorbidities: atopic

dermatitis, allergic endpoints, and autoimmune diseases (Supplementary Data 5 & 10). Atopic dermatitis had probable colocalization (posterior probability > 0.9) with 7 loci, of which 6 had a shared asthma/CRS impact, and one unassigned. Allergic phenotypes colocalized with 16 loci, mostly having a shared asthma/CRS impact, with the exception of 3p21.3 locus harboring *EEFSEC*, which was asthma-specific. Autoimmune diseases colocalized with five loci (5p13.2, 11q13.5, 12q13.2, 12q13.3, and 17q21.32), all with shared asthma/CRS impact. For three of these five loci (5p13.2, 12q13.3, and 17q21.32) the nearest genes (*IL7R*, *STAT6*, and *TBX21*, respectively) have previously reported Mendelian immune deficiency variants. In total, 10 loci harbored genes with such reported Mendelian immune deficiency variants 27 (Supplementary Data 5). These loci with immune deficiency-linked genes shared impact on both asthma and CRS in 9 of 10 loci and one, 10p15.1, remained unassigned, with GWS impact to asthma ($OR_{asthma} = 0.804$ [95% CI 0.756–0.855], $p = 1.90E-12$).

“We also investigated asthma subtype association among asthma-associated loci (Supplementary Data 5 & 10). Here, we colocalized all 117 asthma-associated loci with subtype analyses from FinnGen R9 PheWEB. Specifically, we investigated colocalization with childhood asthma, defined as onset before the age of 16 ($n = 5,865$), and asthma-related infections ($n = 16,018$). Childhood onset was colocalized ($PP > 0.9$) with 10 loci, all with shared impact to CRS, and 8 colocalized with allergic diseases. Asthma-related infections were colocalized ($PP > 0.9$) with 10 shared loci and one unassigned (17q21.33).”

Rows: 217-222: In line with comment above, for the shared associated loci with asthma related sub-groups and phenotypes (such as childhood asthma, autoimmunity, asthma related infections) did you take into account the risk allele and effect size or just genome wide significant p-value? If not the comparison should take into account effect size and risk allele and if opposite allele is associated with similar effect size.

- We hope that we have understood the reviewer’s comment correctly, and assume this relates to the unreported issue of contra-directional impact. As stated above, we have now elaborated this analysis to formal colocalization, which takes into account several confounding factors, including potential differences in the direction of effect between risk allele and reference allele.

Rows 240- : In the section “in silico analyses” the authors describe the MAGMA analysis and results. 608 genes were identified from the different phenotypes, however, the authors don’t describe the input of the analysis. Which genes and SNPs were used for the analysis?

- We thank the Reviewer for this feedback, and have raised the following points from the Methods section to the Results section: “Genes were downloaded from NCBI RefSeq ($n = 195,26$). For gene set analysis, we used canonical gene sets ($n = 7561$) described at MSigDB (v2023.1.Hs).” We have further added the following note to describe the purpose of the MAGMA analysis:
 - o Lines 263-264: This analysis summarizes GWAS results over gene regions to identify associated genes rather than single alleles.

Row 273: “Asthma and CRS are common diseases with often lifelong burden and about half of cases overlap”. The “d” is missing in “and”. Also, the fact that half the cases have both diseases only seem true for CRS cases? Or is there a massive number of undiagnosed CRS individuals among the asthma cases in this study?

- Thank you for noticing the typo, along with the ambiguity of the statement. The leading sentence of the Discussion is now amended to read: “Asthma and CRS are common diseases with often lifelong burden and about half of CRS cases have asthma, possibly due to shared pathomechanisms”.

Row 381: “The co-occurrence of phenotypes is summarized in Supplementary Data 7.”

Supplementary Data 7 is something else. The information is in Supplementary Table 6 and 7.

- Thank you for catching this! We have amended accordingly.

Row 413: Please specify “incomplete minimum phenotype information”?

- We have added the following insert to clarify: “(age, sex, and hospital record data)”

Row 440: “all cases with asthma, CRSwNP or CRSwNP as case.” Should it be “CRSsNP”?

- Thank you for catching this! It should indeed be CRSsNP, we have amended accordingly.

Row 449: **Bayesian analyses:** How did the authors account for possible opposite direction in association between the different phenotypes? Perhaps there were none?

- This is an excellent point! In the Bayesian pipeline itself, the chosen models would simply presume independent signals (a separate model). The choice of focusing on shared impact, specifically the choice of a single set of controls, would here mean that a variant with opposing impact would not show up in the original analysis and would not be considered in the Bayesian analyses – hence, the independent model would be unnecessary here, but is important to point out.

We have added the following to Methods (lines 502–503): “No independent signal model was considered, as the choice of using shared controls would deflate signals from variants with opposing effects.”

We have also added the following note to limitations: “it is therefore likely that further disease-specific **or opposite-effect** variants remain to be identified”

In table 2, risk allele frequency would add to the understanding.

- Thank you! We have added effect allele frequency to Table 2, along with a note in the table legend: “EAF: Effect allele frequency (%) in FinnGen R9.” The allele frequencies (in FinnGen) have also been added to Tables 3 and 4.

Supplementary Data 1 and 2 has a heading “DIRECTION”, why not just write the “risk allele”? It is less confusing (+ and – is also used for strand of Gene expression). Direction is also shown in “BETA” (assuming that the “REF” allele is the one always compared with)

- We thank the Reviewer for highlighting the ambiguity here and have made the following edits to clarify:

The supplementary Data 1 legend now reads “DIRECTION: Codirection between FinnGen and UKB”.

Reviewer #3 comments:

This large study tackles a well-constructed hypothesis using appropriate methodology and has some novel findings, e.g. that CRSwNP specific loci does not associate with allergic phenotypes, and novel non-synonymous associations.

My biggest comment is the lack of results presented between genetic overlap with CRsNP (with both asthma and CRSwNP) in the Bayesian grouping, despite the novel loci being detected including CRSsNP in the case definition. (To not distract from the main results, it may be most appropriate to add these results to the supplementary material.) There are fewer genome-wide significant loci with the CRSsNP specific GWAS, despite larger sample size (probably due to less heritability), and contrasting genetic overlap with CRSsNP would likely be a useful null comparison, providing stronger support for the overlap between asthma and CRSwNP.

- We thank the Reviewer for raising this concern. It was indeed felt that the CRSsNP analysis was too incomplete in UKB to fully explore (as noted in the limitations). However, with this renewed interest, we have added a supplementary figures 1, 3 and 4 on the CRSsNP analysis, and the following notes in manuscript:
 - o Line 133: “For CRSsNP, we identified five GWS loci (**Supplementary Figure 2, Supplementary Data 3**), [...]”
 - o Lines 168–169: “Interestingly, while lacking CRSwNP impact, these three loci showed possible shared effect to CRSsNP (Supplementary Figure 2), and 2q24.1 in particular”.
 - o Lines 182–184: “Considering CRSsNP as well (Supplementary Figure 3), a probable asthma-specific effect (PP > 50 % for ASTHMA model) was apparent for only 6 of 43 loci: 4p14, 4q24, 7p21.1, 8p23.1b, 8p23.1c, and 10p12.31.”

My remaining comments are mostly editorial/requests for clarification.

Specific comments – Abstract

- Lines 44-45: “These loci largely cluster into three groups, with a shared risk being the most frequent.” – I am not sure what this means, or what result this refers to (perhaps the metABF groups?).
 - o Thanks for pointing this out, we agree that this is too vague. This sentence has been removed.

Specific comments – Introduction

- Line 62-63: “CRSwNP is underdiagnosed in the general population, and the true prevalence is likely higher”– is it possible to give a citation for this statement?
 - o We have added here a citation to Johansson et al, and Fokkens et al (European Position Paper on CRSwNP)
- Line 67: “The majority of Western CRSsNP and CRSwNP cases” – perhaps rephrase as “The majority of CRSsNP and CRSwNP cases in Western countries”
 - o We have amended accordingly.
- Lines 95-96: “Our previous in silico analyses using LD score regression [17] established a genetic correlation of 68.7% between asthma and CRSwNP” – ref 17 is the LDscore regression paper; what is the reference for the previous in silico analyses?
 - o Thanks for catching this! We apologize for the oversight, and have amended accordingly:
 - o Lines 95–97: “Our previous in silico analyses[12] using LD score regression[19] established a genetic correlation of 68.7% between asthma and CRSwNP”

Specific comments – Results

- General comment (which pertains to missing information in the methods section but I was curious about this as I read the results) – how did the authors decide when an association was novel vs known? Did they e.g. use GWAS catalog, or literature? If the former, which search terms did they use? Did they match SNPs exactly, or did they allow for a window around SNPs?
 - o This is a good point, also raised by reviewer 1. We have added a note on the novelty as follows (lines 473–476):

“We defined loci as novel if they did not appear in OpenTargets or GWAS Catalog with an association to the associated phenotype. We also manually curated loci according to the phenotypes of asthma[17,18,23,24,67], CRSwNP[9,12] or CRSsNP[12].”
- Lines 114-115: “GWAS results from FinnGen and UKB were meta-analyzed for validity and replication” – This wording does not make sense to me (this is not replication in the traditional sense where there is a very clear discovery and replication cohort) – do you rather mean to assess heterogeneity of effects between cohorts?
 - o Thanks for noticing the vague nature of the statement. We agree that the use of the word “replication” here is inappropriate, and have removed it. Note that as per Reviewer 1’s request, this sentence has been moved to Methods (line 469).
- Lines 120-121: “While four asthma lead variants had a heterogeneous effect estimate ($I^2 > 50\%$)” – in table 2, only 2 variants had $I^2 > 50\%$ (rs114425738-G and rs11168249-C), not 4 variants as stated.
 - o Apologies, the two missing heterogeneous loci had been detected not to be novel and had been removed from the table, but not corrected here prior to initial submission. We have updated accordingly, and thank reviewer 3 for catching this!

With comments from reviewer 1, we have now removed a further 7 loci as they have been reported before, so there is only one locus (1p22.1) with heterogeneous effect. Hence, this section now reads (lines 117–119): “All novel asthma lead variants had co-directional impact with at least nominal significance ($p < 0.05$) in both cohorts, although heterogeneity for 1p22.1 (harboring *TGFBR3*) was high ($I^2 = 76.3\%$).”
- Lines 121-122: Weird line break
 - o Thanks for catching this typo, it has been removed
- Lines 132-137: From the wording, it is not clear which one of the associations are novel
 - o Thanks, we have revised this accordingly:

“For CRSsNP, we identified five GWS loci (Supplementary Figure 2, Supplementary Data 3), including one novel association at 9q33.3, near *NEK6*. The locus, previously linked to allergic rhinitis and vitiligo, associated with CRSsNP (OR = 1.069 [95 % CI 1.044–1.094], $p = 2.28 \times 10^{-8}$).”
- Line 149: “However, as the intercept values” should be “However, as the LD score regression intercept values”
 - o Thanks for pointing this out, this is indeed too unspecific. We have amended as suggested.
- Line 152: “the expected associations” should be “the expected distribution”
 - o Thanks for catching this, we have amended as suggested.
- Lines 155-160: It is probably also worth pointing out that the other advantage of using MetABF is the ability to compare effects across phenotypes with varied sample size
 - o We thank Reviewer 3 for appreciating this! We have added a note on this as suggested (lines 159–161):

“This method utilizes a Bayesian tree analysis framework to identify the most probable underlying model from a given set while accounting for overlapping cases, in addition to prevalence differences.”
- Lines 183-193: While I agree that this is an interesting observation that should be pointed out, I think it should be attenuated somewhat: While the CRSwNP point estimates are mostly higher in Figure 5, the confidence intervals of most overlap with asthma. In addition, the closing sentence noting the disease-specific overlap in findings are based on loci that crossed genome-wide significance, which is limited by the fact that the disease-specific GWAS are of different sample size.

- We thank the Reviewer for raising this concern, which also attunes with a concern raised by Reviewer 2. We have made the following edits:

Lines 198–199: “with a slope of 1.88 in favor of CRSwNP in log-odds space, **even if 95 % confidence intervals of individual loci still overlapped with risk for asthma**”

Lines 203–207: “In our disease-specific GWASs, 22.9 % (20/87) of asthma loci were determined to not impact CRSwNP. **Similarly, 28 % (14/50) of CRSwNP loci exhibited no significant impact on asthma despite the significantly lower case count of CRSwNP, further emphasizing the nuanced genetic landscape shared between these conditions.**”

- Characterization of loci section: Suggest splitting it out into two separate sections, one focused on related phenotype associations, and one focused on fine-mapping
 - We thank the Reviewer for this suggestion, we feel it significantly improves the readability. We have amended accordingly, replacing Characterization of loci with “**Colocalization analyses**” and adding a “**Fine-mapping of potential causal variant**”
- Line 226: Suggest dropping “considered” (either the association crossed the threshold, or not)
 - Thanks for pointing this out, the sentence now reads:” These results were meta-analyzed, **with GWS variants reported in Table 6.**” (Lines now 212-213).
- Line 256-258: Could this observation perhaps be due to underrepresentation of asthma relevant cells in the GTEx lung tissue?
 - We appreciate this highlight, and have added “**This analysis is limited due to underrepresentation of airway tissues.**” to line 284.
- Line 266: “overlapped 100%” – it was not clear to me what this meant – perhaps restate this as the confidence interval upper bound reaching 100%. In addition, for supplementary table 3 etc, correlation coefficients should be expressed as a proportion, not a percentage
 - Thanks, we have amended as suggested (lines 292–293): “Genetic correlation between FinnGen and UKB analyses **has a confidence interval upper bound of 100 % for CRSwNP (p = 1.66E-29), [...].**”
- Line 270-271: A couple of brief sentences stating the previous method that was used, and why it led to inflated estimates, would be appreciated.
 - Indeed, we have elaborated on this point (lines 296–300):

“**In our previous study, SAIGE was used for genome-wide association, which is based on saddlepoint approximation and has been reported to inflate effect estimates. This has likely also inflated our previous genetic correlation estimate, and we hold our present estimates based on REGENIE genomic analysis to be more reliable.**”

Specific comments – Discussion

- Line 273: The grammar of the opening sentence can be improved.
 - Thanks for catching this, which was also noted by Reviewer 2. The opening sentence now reads (lines 302–303):

“**Asthma and CRS are common diseases with often lifelong burden and about half of CRS cases have asthma[37], possibly due to shared pathomechanisms [38,39].**”
- Line 278: Suggest describing ancestry as European ancestry rather than Caucasian (PMID: 35212875)
 - Agreed, and amended as suggested
- Line 299: Suggest “differ from” instead of “distinguish from”
 - Agreed, and amended as suggested.
- Line 353: “suggest inflation likely” should be “suggest inflation is likely”
 - Corrected

Specific comments – Materials and Methods

- Lines 375-380: The phenotype definitions are more broad than the criteria stated on lines 110-112; suggest updating lines 110-112 accordingly
 - This is a good point. Given previous comments to the paragraph in question, the partial phenotype definitions provided in the results are now seen as unnecessary and misleading, and have been removed. The lines 113–115 now read:

“We performed genome-wide association studies (GWASs) of asthma, chronic rhinosinusitis with nasal polyposis (CRSwNP), and chronic rhinosinusitis without nasal polyposis (CRSsNP) in FinnGen and UKB (Table 1, Figure 1).”
- Line 381: Supplementary Data 7 seems to be an author list
 - Thanks for catching this, we have amended this to read “Supplementary **Tables 6 and 7**”
- Line 440: “CRSwNP or CRSwNP” should be “CRSwNP or CRSsNP”
 - Thanks for catching this typo, we have corrected as suggested.
- Lines 503-506: MetABF software details are missing
 - Indeed, thanks for pointing this out. The corresponding lines 555–557 now read “The Bayesian analysis frameworks ~~is~~ are publicly detailed in the cited work[**26,29**], in addition to their **respective** websites (**<https://github.com/trochet/metabf>**; **<https://github.com/mjpirinen/linemodels>**).

Response to reviewer comments

Reviewer #1 comments:

The authors have conducted additional analyses and revised the manuscript to my satisfaction. One recommendation would be to edit Figure 1 so that it is more illustrative; consider making the points larger and including breaks in the y-axis to reduce the white space.

- We thank Reviewer #1 for their valuable insights. We have revised the Miami plot by adjusting the scale with larger points to reduce whitespace.

Reviewer #2 comments:

I am pleased with the authors' response and commend their thorough investigation of GWAS and co-morbidity, which presents a significant challenge.

- We thank Reviewer #2 for their efforts, which have allowed us to clarify the description of our study and broadening its potential audience.

Reviewer #3 comments:

No additional comments; the revision adequately addressed my previous comments.

- We thank Reviewer #3 for their time and contributions to the furthering of this manuscript.

Reviewer report on Manuscript#: NCOMMS-24-57600-T

Reviewer: Åsa Torinsson Naluai, Associate Professor at the Sahlgrenska Academy at the University of Gothenburg Sweden.

In the manuscript titled: “131 genetic loci highlight immunological pathways and tissues in nasal polyposis and asthma” Elmo Saarentaus and coworkers show for the first time a genome-wide meta-analysis of asthma and CRSwNP in a very large number of clinical samples from the Finnish and British populations. They detect 17 new loci associating with asthma, 33 new loci associating with CRSwNP, and one locus associating with CRSsNP. Comparing effect estimates allows the authors distinguish a total of 131 loci, of which 20 associate only with asthma, 14 only with CRSwNP, and 71 loci with both.

Elmo Saarentaus and coworkers adopted an approach of combining cases from the three disorders (asthma, CRSsNP and CRSwNP) to emphasize shared biology, and shared genetic impact, more than distinct genetic impact, and suggests that most shared-impact loci also appear in the original asthma or CRSwNP-specific GWASs performed in this study.

In summary,

- What are the noteworthy results?

The authors distinguish a total of 131 loci, of which 20 associate only with asthma, 14 only with CRSwNP, and 71 loci with both. Noteworthy is the overlap, which might also be overestimated.

- Will the work be of significance to the field and related fields? How does it compare to the established literature? If the work is not original, please provide relevant references.

Yes, this work will add significant knowledge to the field.

- Does the work support the conclusions and claims, or is additional evidence needed?

Some conclusions are supported by the data while some might be slightly different if the analysis was done differently.

- Are there any flaws in the data analysis, interpretation and conclusions? Do these prohibit publication or require revision?

The study focusses on shared-impact loci and has been performed very well, but I suggest the original asthma or CRSwNP-specific GWASs performed in this study are done differently.

- Is the methodology sound? Does the work meet the expected standards in your field?

Yes

- Is there enough detail provided in the methods for the work to be reproduced?

Some additional details in certain sections have been asked for in comments below.

Major comments:

One of the primary goals of this paper is to identify shared genetic factors and effects where the influence is either identical or very similar for both asthma and CRSwNP, as well as to pinpoint distinct effects where the genes or effects are unique to each condition. On row 356, the authors summarize the study's findings by stating: "*On the whole, this study shows that shared heritable affinity to developing CRSwNP has an even higher overlap with that of asthma than expected*". However, the authors do not separate the patients in the GWASs into distinct groups of those with asthma only and those with CRSwNP only. Instead, they include individuals with asthma in the CRSwNP GWAS analysis and those with CRSwNP in the asthma GWAS analysis. As shown in Figure 1: Study Overview and in Supplementary Figures 6 and 7, the GWAS of CRSwNP includes nearly 40% of individuals with asthma, and nearly 30% of those diagnosed with CRSsNP also have asthma (data from FinnGen only, as discussed below).

To more accurately identify genes specifically associated with CRS, I believe it would be preferable to conduct a GWAS on individuals diagnosed exclusively with CRS, as well as a GWAS on those with asthma only. The patients who have both CRSwNP and asthma will have genes linked to both conditions, and for this effect, the authors already have the combined GWAS data from 88,965 individuals. If the authors suggest that a genetic effect is seen for asthma but not for CRSwNP, or vice versa, shouldn't the asthma cases be excluded from the CRSwNP analysis, and vice versa? The findings and the identified disease-specific GWAS results—where 22.9% (20/87) of asthma loci were considered not to influence CRSwNP, and 28% (14/50) of CRSwNP loci were found not to significantly affect asthma—will likely change if the GWAS only include patients with one condition or the other, rather than both. This study also have more samples and the power lost will most likely be replaced by more specific results for each disease.

It is also noted in these tables that CRSsNP does not overlap with asthma diagnosis in the UK Biobank (UKB) samples, but there is a 30% overlap in FinnGen. This difference should perhaps be addressed earlier in the manuscript. In the discussion, the authors state, "CRSsNP also appeared underrepresented in the UKB, underpowering its analysis considerably." Would it be more accurate to say something like: "The analysis of CRSsNP in the UKB dataset was not possible because the ICD code for CRSsNP was not available in the population"?

Minor comments:

Rows 186-188 the authors write: "*Notably, for loci associating with both asthma and CRSwNP, the effect on CRSwNP was consistently higher than that of asthma, with a slope of 1.88 in favor of CRSwNP in log-odds space.*" Isn't this just due to the fact that the CRSwNP sample size is smaller and that you only include genome-wide significant SNPs in the results?

In the section "**Characterization of loci**" the authors seem to be referring to specific associated SNPs or loci in general (and not whole genes). If by "loci" you mean "SNP" please specify this and if possible, include the rs-numbers and risk alleles together with each gene name mentioned.

In some places the authors write “*nearest gene*” and in some places just “*gene*” (also in supplementary data). Please specify if it is always nearest gene and if not specify why. Also in supplementary data 6, some genes are written as Gene ID and other as Ensemble ID.

Corresponding NCBI RefSeq gene name would be good to include (since these gene names are used in every other table). It also would be informative to include top associated SNP in Supplementary Data 6.

Row 177: “*epidemiological overlap*” is not completely clear what is meant. Could the authors please specify.

Rows 206- The authors write “shared impact on CRSwNP and asthma were more often co-directional with other inflammatory endpoints. Atopic dermatitis had co-directional impact with 18.6 % (13/70) of shared loci.” Please specify if you only count genome-wide associated results or if you look at all SNPs in each case. It would be interesting to compare all of your 70 SNPs in which direction it is, if this was not done?

Rows: 217-222: In line with comment above, for the shared associated loci with asthma related sub-groups and phenotypes (such as childhood asthma, autoimmunity, asthma related infections) did you take into account the risk allele and effect size or just genome wide significant p-value? If not the comparison should take into account effect size and risk allele and if opposite allele is associated with similar effect size.

Rows 240- : In the section “**in silico analyses**” the authors describe the MAGMA analysis and results. 608 genes were identified from the different phenotypes, however, the authors don’t describe the input of the analysis. Which genes and SNPs were used for the analysis?

Row 273: “*Asthma and CRS are common diseases with often lifelong burden and about half of cases overlap*”. The “d” is missing in “and”. Also, the fact that half the cases have both diseases only seem true for CRS cases? Or is there a massive number of undiagnosed CRS individuals among the asthma cases in this study?

Row 381: “*The co-occurrence of phenotypes is summarized in Supplementary Data 7.*” Supplementary Data 7 is something else. The information is in Supplementary Table 6 and 7.

Row 413: Please specify “*incomplete minimum phenotype information*”?

Row 440: “*all cases with asthma, CRSwNP or CRSwNP as case.*” Should it be “CRSsNP”?

Row 449: **Bayesian analyses:** How did the authors account for possible opposite direction in association between the different phenotypes? Perhaps there were none?

In table 2, risk allele frequency would add to the understanding.

Supplementary Data 1 and 2 has a heading “DIRECTION”, why not just write the “risk allele”? It is less confusing (+ and – is also used for strand of Gene expression). Direction is also shown in “BETA” (assuming that the “REF” allele is the one always compared with).